# Rethinking Knowledge Graph Evaluation Under the Open-World Assumption

**Haotong Yang**[12]    **Zhouchen Lin**[123*]   **Muhan Zhang**[24*]
[1]Key Lab of Machine Perception (MoE),
School of Intelligence Science and Technology, Peking University
[2]Institute for Artificial Intelligence, Peking University
[3]Peng Cheng Laboratory
[4]Beijing Institute for General Artificial Intelligence
{haotongyang, zlin, muhan}@pku.edu.cn

## Abstract

Most knowledge graphs (KGs) are incomplete, which motivates one important research topic on automatically complementing knowledge graphs. However, evaluation of knowledge graph completion (KGC) models often ignores the incompleteness—facts in the test set are ranked against all unknown triplets which may contain a large number of missing facts not included in the KG yet. Treating all unknown triplets as false is called the closed-world assumption. This closed-world assumption might negatively affect the fairness and consistency of the evaluation metrics. In this paper, we study KGC evaluation under a more realistic setting, namely the *open-world assumption*, where unknown triplets are considered to include many missing facts not included in the training or test sets. For the currently most used metrics such as mean reciprocal rank (MRR) and Hits@K, we point out that their behavior may be unexpected under the open-world assumption. Specifically, with not many missing facts, their numbers show a logarithmic trend with respect to the true strength of the model, and thus, the metric increase could be insignificant in terms of reflecting the true model improvement. Further, considering the variance, we show that the degradation in the reported numbers may result in incorrect comparisons between different models, where stronger models may have lower metric numbers. We validate the phenomenon both theoretically and experimentally. Finally, we suggest possible causes and solutions for this problem. Our code and data are available at https://github.com/GraphPKU/Open-World-KG.

## 1  Introduction

Knowledge graph (KG) is a structural method to store facts about some field or the world. Because most KGs are incomplete, the knowledge graph completion (KGC) task is proposed to automatically complement the existing KG with missing facts. However, when we do not know the missing facts in advance, we must manually evaluate whether each predicted completion is correct, which is an impossible task for modern KGs. This problem is called the *open-world problem* and the assumption that KGs are incomplete is called the *open-world assumption*. A general solution is to extract the training, validation and test sets from the existing incomplete KG and then evaluate the trained models on the test set. Then, a natural question is whether the conclusion drawn from the incomplete test set is consistent with the true strength of the model, which should be measured on the complete KG.

To answer this question, we need to investigate the metrics used to evaluate KGC models. KGC models are often evaluated by ranking-based metrics, such as mean reciprocal rank (MRR) and

---

*Corresponding authors.

36th Conference on Neural Information Processing Systems (NeurIPS 2022).

Table 1: The filtered ranking as well as the metric MRR. "w/o c": without correction and "with c": with correction. Query: "*What sports were included in the 1956 Summer Olympics?*"

| | Test Answers | | Missing Answers | | | | | |
|---|---|---|---|---|---|---|---|---|
| | swimming | sailing | **water polo** | **boxing** | **dressage** | **show jumping** | **canoe sprint** | **cycling** |
| ranking w/o c | 5 | 5 | 1 | 2 | 3 | 4 | 7 | 9 |
| ranking with c | 1 | 1 | 1 | 1 | 1 | 1 | 3 | 4 |

Hits@K. Under the open-world assumption, when a missing fact that should have been included in the test answers is predicted by the model, **its ranking could be higher than some test answers, which makes the rankings of these test answers drop**. In this situation, despite actually recognizing more right answers, the metrics drop instead.

To intuitively show the problem, we train BetaE [Ren and Leskovec, 2020], one state-of-the-art multi-hop KGC model, on the FB15k-237 dataset [Toutanova and Chen, 2015]. One of the test queries is "*What sports were included in the 1956 Summer Olympics?*". The two test answers are swimming and sailing, both with the filtered rankings (refer to Section 2) of 5, so the MRR on this query is 20%. However, when we manually check the first 30 predictions , we found that many of them are in fact sports included in the 1956 Summer Olympics but not included in the answer set.[2] We present these missing answers in Table 1. We can see that all the four sports previously ranking higher than the two test answers turn out to be missing true answers. Thus, if we correct the answer set by adding these missing answers to the test set, the actual filtered rankings of the two test answers are both 1, and the corrected MRR on the new test set becomes 82% which is much higher than the reported 20%, indicating that the model strength on this query is significantly underestimated.

In this paper, we study the odd behavior of the ranking-based metrics under the open-world assumption, and summarize two problems affecting the KGC evaluation: 1) **Metric Degradation**. It means that with the increasing of the actual model strength, the increasing of the reported metric becomes slower and slower. Thus, the reported metric might not be able to reflect the true model improvement. 2) **Metric Inconsistency**. It means that when comparing two models, the model with lower reported metric may actually have better performance if we evaluate them on the complete KG.

Our main contributions include that: For the first time, we theoretically analyze the evaluation of KGC under the open-world assumption and point out the degradation and inconsistency problems. Furthermore, we suggest that the degradation and inconsistency may be related to the focus-on-top behavior of the metrics, and provide a solution to relieve the two problems. Finally, we verify the theoretical analysis through experiments on an artificial closed-world KG.

## 2   Background and related work

**Knowledge graph completion**   Current KGC models can be mainly categorized into three classes: logic-based, embedding-based, and neural-based. **Logic-based models** [Joseph and Riley, 1998, Richardson and Domingos, 2006] use some explicit rules for KGC, which are manually provided or mined by some rule-mining methods, such as [Galárraga et al., 2013, Yang et al., 2017, Sadeghian et al., 2019]. These models search through the existing KG and deduce missing facts according to the given rules. However, this process can be time-consuming and noise-sensitive. At the same time, if the KGs are highly incomplete, the performance could be poor. **Embedding-based models** [Bordes et al., 2013, Yang et al., 2015, Trouillon et al., 2016, Sun et al., 2019] represent entities and relations by learned vectors or tensors, where the possibility of a fact is measured by a *score function*. These models have good scalability and can be applied to large and sparse KGs. Some works aim to generalize embedding-based models to more patterns [Trouillon et al., 2016, Abboud et al., 2020] and more assumptions (such as multiple answers) [Vilnis et al., 2018, Ren et al., 2020, Abboud et al., 2020]. One of the interesting directions is to consider multi-hop reasoning [Ren et al., 2020, Ren and Leskovec, 2020, Zhang et al., 2021], where a query can be composed by several conditions, such as "*Who is the Canadian and won the Turing Award ?*" Note that the open-world problem could be more severe in the multi-hop setting, because missing in any condition leads to missing in the final results. **Neural-based models** combine neural networks with embeddings. Dettmers et al. [2018] and Nguyen et al. [2018] use a convolution networks as the score function to enlarge the capacity of

---

[2]All sports held at this Olympics are in: `https://en.wikipedia.org/wiki/1956_Summer_Olympics`.

the models. Nathani et al. [2019], Vashishth et al. [2020] and Wang et al. [2021] use graph neural networks on the KGs to learn the embeddings or directly predict the links.

**KG evaluation**    Current KGC evaluation resorts to manually split training, validation and test sets from the incomplete KG. Given a test query $r(e_h, ?)$ (which entities have the relation $r$ with the head entity $e_h$?), a typical method is to predict a score for all entities as the tail entity, rank all the entities, and then measure the average of a ranking-based function $h(\boldsymbol{r})$ on the test answers. Here, the most-used metrics are MRR $h(\boldsymbol{r}) = 1/\boldsymbol{r}$ and the Hits@K $h(\boldsymbol{r}) = \mathbb{I}(\boldsymbol{r} \leq K)$. Because there could be multiple answers for a query, the metrics should be **filtered**, which means the answers in the training and test sets do not occupy a position so that the number of training and test answers does not affect the metrics. The details of the filtering can be found in [Bordes et al., 2013]. Due to the nonlinearity of most ranking-based metrics, some works have theoretically investigated their behavior. Wang et al. [2013] point out some ranking-based metrics always converge to 1 on different models as the number of objects to rank goes to infinity, so that the performance of models is indistinguishable. Krichene and Rendle [2020] analyze the behavior of ranking-based metrics under negative sampling. They point out the sampled metrics can be inconsistent with exact metrics and all metrics lose their focus-on-top feature and collapse to a linear one, AUC-ROC, in the small sample limit. Sun et al. [2020] focus on the unfair tie-breaking methods. Akrami et al. [2020] find some data argumentation such as adding inverse relations could be a kind of excessive data leakage during evaluation.

**Open-world assumption**    Some recent works have noticed the gap between the actual open-world situation and the closed-world assumption. Cao et al. [2021] point out that the closed-world assumption leads to a trivial evaluation on the triple classification task. They offer their manually-labeled positive-negative-unknown ternary triple classification datasets following the open-world assumption and point out the lack of capacity for current models to distinguish *unknown* from *negative*. However, the *unknown* part in the dataset is only on the triple classification task, while we focus on the link prediction task here. Additionally, Das et al. [2020] analyze the open-world setting as an evolving world that continuously adds new entities and facts to KGs. Under this setting, their work focuses on the *inductive* or *case learning* capacity, i.e., the capacity of models to generalize on unobserved entities. Here, we aim to analyze the possible inconsistent comparison in evaluation with missing facts instead of a specific framework with a larger *inductive* capacity.

## 3    Open-world problem

In this section, we formally define the open-world problem that will be analyzed in our paper.

**Definition 3.1** (Knowledge Graph). A *knowledge graph* is a relational graph $G = (E, F, R)$ where $E$ is the vertex set containing *entities*, $F$ is the edge set containing *facts*, and $R$ is the *relation* set. Each edge $f \in F$ is labeled by a relation. If an edge $f$ between entities $e_h$ and $e_t$ is labeled by relation $r \in R$, we denote the edge $f$ as $r(e_h, e_t)$ where $e_h$ is the head entity and $e_t$ is the tail entity.

In this paper, we assume $E$ and $R$ are fixed. Therefore, we sometimes directly use $G$ to denote the fact set $F$, and $r(e_h, e_t) \in G$ means there is relation $r$ between $e_h$ and $e_t$ in the KG $G$.

A set of KGs with the same entities $E$ and relations $R$ but different facts $F$ is called a *world* and denoted by $W(E, R)$. An (open-world) KG can be considered as an observation or understanding of the world where there could be unobserved or unknown facts, while the closed-world KG contains all the true facts of the world. Formally, the closed-world and open-world KGs are defined as follows:

**Definition 3.2** (Closed-World KG and Open-World KG). For a world $W(E, R)$, the closed-world KG $G$ is the closure of the world.
$$G = \bigcup_{G' \in W} G',$$
where the union is defined on the fact set. And for a KG $G' \in W$, if $G' \neq G$, $G'$ is open-world.[3]

Some property of closed-world KGs: 1) There is a one-to-one correspondence between closed-world KGs and worlds $W(E, R)$. 2) All the KGs in a world are subgraphs of the closed-world one. 3) If $G$ is the closed-world KG of the world $W$, we have $f \notin G \Rightarrow \forall G' \in W, f \notin G'$.

---

[3]Some works use open-world to refer to not only facts but also entities may be incomplete.

The third property is critical. It means with the closed-world $G$, we **know** there is **no such a relation** $r$ between $e_h$ and $e_t$ in the world when $r(e_h, e_t) \notin G$. Given the closed-world KG, we have all knowledge of the world, including both the *positive* and *negative* one. Conversely, if a KG is open-world, we **do not know** whether the triplet is *false* or *unknown* when $r(e_h, e_t) \notin G$. In other words, an open-world KG only contains *positive* knowledge.

In the rest of the paper, we denote the closed-world KG as $G_{full}$. Because we want to study the evaluation, we denote the existing open-world dataset as $G_{test}$, and extract the training set $G_{train}$ from $G_{test}$. Here, $G_{train} \subseteq G_{test} \subseteq G_{full}$ and the facts in $G_{train}$, $G_{test} \setminus G_{train}$, $G_{full} \setminus G_{test}$ are **training facts**, **test facts** and **missing facts** respectively. In addition, we also call the facts in $G_{full} \setminus G_{train}$ **full test facts** and $G_{test} \setminus G_{train}$ **sparse test facts**.

Now, we can formally define the *open-world problem*. We believe the actual strength of a model should be evaluated on the full test facts $G_{full} \setminus G_{train}$. However, because the closed-world KG $G_{full}$ is unavailable, the evaluation is often performed over $G_{test} \setminus G_{train}$. The question is:

> *Whether the conclusions from evaluation on the sparse test facts $G_{test} \setminus G_{train}$*
> *lead to consistent conclusions from evaluation on the full test facts $G_{full} \setminus G_{train}$.*

## 4 Theoretical analysis on metric degradation and inconsistency

To study the open-world problem, we theoretically analyze the behavior of ranking-based metrics with missing facts. All the proofs are in Appendix A.1. The randomness comes from two sources: the missing of facts and the predictions of the model. We model them as two random events.

- Missing Fact Model: For a full test fact $r(e_h, e_t) \in G_{full} \setminus G_{train}$, $X$ means it is a missing fact with $P(X) = \beta$ while $\overline{X}$ means it is included in the sparse test set $G_{test} \setminus G_{train}$ with $P(\overline{X}) = 1 - \beta = \alpha$. $\beta$ is called the sparsity of the KG.

- Prediction Model: For simplicity of analysis, we model KGC as a classification task. In fact, an ideal (oracle) KGC model is exactly a classification model, which identifies all the correct facts. Here, for a full test fact $r(e_h, e_t) \in G_{full} \setminus G_{train}$, $Y$ means the answer $e_t$ is correctly classified as positive with $P(Y) = \ell$. $\ell$ is called the strength of a model. We break ties uniformly at random for entities classified into the same class.

Note that one of our basic assumptions is *'set answer'* followed [Ren et al., 2020], where we believe the answer for a query should be a *set* or *concept* instead of exactly one entity. For the open-world problems we care about, we assume that the number of elements in the answer set should be larger than one.

### 4.1 Expectation degradation

We first assume the independence of the random events $X$ and $Y$. We show that the expectation of the metrics will degrade with missing facts. Specifically, the increasing of the metrics shows a logarithmic trend, so that it could be too flat to reflect the true increasing of the model strength.

Assume the number of entities is $N_{entity}$ in the KG. For a given query $r(e_h, ?)$, let $N$ be the number of full test answers. The random variant $\boldsymbol{m}$ is the number of missing answers $G_{full} \setminus G_{test}$, and it follows the binomial distribution $\mathcal{B}(N, \beta)$. The other $N - \boldsymbol{m}$ answers are test answers. We denote the filtered ranking of the entity $e$ as $\boldsymbol{r}(e)$. Then we have the lemma.

**Lemma 4.1** (Expectation of ranking-based metrics)**.** *With the modeling of missing fact and prediction as above, the expectation of ranking-based metric $\mathfrak{M} = \frac{1}{N - \boldsymbol{m}} \sum_{i=1}^{N-\boldsymbol{m}} \frac{1}{f(\boldsymbol{r}(e_i))}$ can be expressed as*

$$\mathbb{E}(\mathfrak{M}) = \frac{1}{\beta(N+1)} \sum_{k=0}^{N} \frac{1}{f(k+1)} \left(1 - \hat{\Phi}(k)\right) + \delta, \tag{1}$$

*where $\hat{\Phi}$ is the cumulative distribution function (cdf) of binomial distribution $\mathcal{B}(N + 1, l\beta)$ and $0 < \delta \leq (1 - l) \frac{\ln(N_{entity} - N)}{N_{entity} - N}$.*

Generally, the $N_{entity}$ is large and the answer rate $N/N_{entity} < 10\%$ in almost all queries, so the item $\delta$ is negligible. In the rest of the paper, we denote $\hat{\mathbb{E}} = \frac{1}{\beta(N+1)} \sum_{k=0}^{N} (1 - \hat{\Phi}(k))/f(k+1)$ which is a good approximation of $\mathbb{E}$.

With this lemma, we get a closed-form expression of the expectation of the ranking-based metric $\mathfrak{M}$. However, this expression cannot explain why the metrics will degrade. Next, we derive the form of derivative of the expectation $\hat{\mathbb{E}}$ w.r.t the model strength $l$ to account for its degradation.

**Corollary 4.1** (Derivative of Expectation w.r.t Model Strength). *Let $g(\boldsymbol{r}) = \boldsymbol{r}/f(\boldsymbol{r}), \boldsymbol{r} \in \mathbb{N}_+$ and $g(0) = 0$. Under the condition of 4.1, the derivative of the expectation w.r.t the model strength $\ell$ is*

$$\frac{\mathrm{d}\hat{\mathbb{E}}(\mathfrak{M})}{\mathrm{d}\ell} = \frac{1}{l\beta(N+1)} \mathbb{E}_{k \sim \mathcal{B}(N+1, \ell\beta)} \, g(k). \tag{2}$$

And for the most-used metrics MRR and Hits@K, their derivative are expressed as follows.

**Corollary 4.2** (Derivative of MRR w.r.t strength $\ell$). *For MRR where $f(\boldsymbol{r}) = \boldsymbol{r}, \forall \boldsymbol{r} \in \mathbb{N}_+$, its derivative w.r.t. $\ell$ is*

$$\frac{\mathrm{d}\hat{\mathbb{E}}(\mathrm{MRR})}{\mathrm{d}\ell} = \frac{1 - \varepsilon}{\ell\beta(N+1)}. \tag{3}$$

*where $\varepsilon = (1 - \ell\beta)^{N+1}$.*

When $\ell\beta$ and $N$ are not too small, the term $\varepsilon$ is negligible. In this situation, the derivative of the metric MRR w.r.t the model strength is approximately of $\mathcal{O}(1/N\ell)$, which will result in insignificant changes in the metric with the increasing of the model strength.

**Corollary 4.3** (Derivative of Hits@K w.r.t strength $\ell$). *For Hits@K where $f = 1$ for $\boldsymbol{r} \leq k$ and $f = +\infty$ otherwise, the derivative is*

$$\frac{\mathrm{d}\hat{\mathbb{E}}(\mathrm{Hits@K})}{\mathrm{d}\ell} = \Phi(K - 1), \tag{4}$$

*where $\Phi$ is the cdf of the binomial distribution $\mathcal{B}(N, \ell\beta)$.*

The behavior of Hits@K is similar to MRR when $K \ll N$. When $\ell\beta$ is not too small and $K$ is not too large, the derivative $\Phi(K - 1)$ is so small that the increase could be insignificant.

Finally, we further approximate Equation (1) by a more intuitive expression with a tolerable error.

**Theorem 4.1** (Expectation of MRR). *For MRR, we can further approximate its expectation by*

$$\hat{\mathbb{E}}(\mathrm{MRR}) \approx \frac{\ln(\ell) + \ln(\beta) + \ln(N + 2) + \gamma}{\beta(N + 1)} := \tilde{\mathbb{E}},$$

*where $\gamma \approx 0.577$ is the Euler's constant and the error $e = |\tilde{\mathbb{E}} - \hat{\mathbb{E}}| \leq \max\{\frac{1}{2\beta(N+1)^2}, \frac{(1-\ell\beta)^{N+1}}{1-(1-\ell\beta)^{N+1}} \cdot \frac{\ln(1/(\ell\beta))}{\beta(N+1)}\}$.*

Firstly, we explain the rationales of the approximation $\tilde{\mathbb{E}}$ as follows.

- $N$ is large enough. For many KGs, especially those commonsense KGs which are not limited to a certain field, the number of answers is often quite large. In the experiment conducted by Ren and Leskovec [2020], there are many tests queries with dozens or hundreds of answers.

- Sparsity $\beta$ is not too small. For most real-world KGs, although we do not exactly know their sparsity, we expect many of them could have a rather high sparsity $\beta$ due to the incompleteness of knowledge extraction and the long-tail distribution of commonsense knowledge.

- Model strength $\ell$ is not too small. Here we are more concerned with how to select and evaluate those models that perform well.

Under the above conditions, the relative error $e$ is negligible. To further show that our approximation is reliable, we do numerical simulations as shown in Figure 1. For $1 - \alpha \geq 0.2$ and $\ell > 0.3$, the analytical and numerical curves almost overlap, which means the *log* approximation is accurate. At the same time, we point out the *log* trend is just the reason why the curve becomes flatter and flatter. The details of the simulation is in Appendix A.2.

Theorem 4.1 illustrates the metric degradation intuitively. There are some conclusions about the MRR under the open-world assumption: 1) Although the theoretical maximum of MRR is 1, the expectation of the MRR of a perfect model $\ell = 1$ is still much lower than 1 and depends on the sparsity of the KG. 2) With the sparsity $\beta$ not very small, the MRR will be a *log* function of the strength $\ell$ times the answer number $N$ which means that as the model gets stronger, the increase of the metric MRR will be less and less significant. The sparser the KG is, the more severe the degradation problem is. Note that in the closed-world KG, the curve should be very closed to $y = x$.

Figure 1: Theoretical *log* approximation (a) and numerical simulation (s). The shadow shows the $[-2\sigma, 2\sigma]$ interval, where $\sigma$ is the numerical std.

### 4.2 Inconsistency due to high variance

In Figure 1, another notable phenomenon is the vibrated curves, which suggests instability of the metric and relatively high variance. This phenomenon combined with the flattening of the expectation can lead to *inconsistency*, which means higher MRR might not mean stronger models unless the difference of the metric is large enough, because the increasing of expectation could be easily overwhelmed by the variance.

One trivial method to solve the problem is to use more test queries. Here we show the number of queries required to ensure the reliability of conclusions can be very large.

**Theorem 4.2** (Consistence with High Probability). *Assuming the number of test queries $N_q$ is large enough ($N_q > 50$), we can approximate the average MRR $\mathfrak{M} = \frac{1}{N_q} \sum_{i=1}^{N_q} \mathrm{MRR}(q_i)$ to follow a normal distribution. Given two independent models $\mathcal{M}_1$ and $\mathcal{M}_2$ whose strength is $\ell$ and $\ell + \Delta\ell$ respectively. The probability of inconsistency between two models can be approximated as follows.*

$$P\left[\mathfrak{M}(\mathcal{M}_1) \geq \mathfrak{M}(\mathcal{M}_2)\right] = \Psi\left(-\frac{\sqrt{N_q}\ln(1 + \frac{\Delta\ell}{\ell})}{\beta(N+1)\sqrt{V(\beta,\ell) + V(\beta,\ell+\Delta\ell)}}\right), \qquad (5)$$

*where $\Psi$ is the cdf of the standard normal distribution.*

Note for a given KG, the sparsity $\beta$ and the answer number $N$ are fixed. Assuming $0 < \frac{\Delta\ell}{\ell} \ll 1$, we have $V(\beta,\ell) \approx V(\beta,\ell+\Delta\ell) := V$ and $\ln(1 + \frac{\Delta\ell}{\ell}) \approx \frac{\Delta\ell}{\ell}$. Then we have the following corollary.

**Corollary 4.4** (Lower Bound of the Number of Queries). *Under the above assumption of $\frac{\Delta\ell}{\ell} \ll 1$, with the upper bound of inconsistency probability $p$, the number of test queries required $N_q$ has a lower-bound as follows.*

$$N_q \geq \frac{c(\beta,\ell,N,p)}{(\Delta\ell)^2}, \qquad (6)$$

*where $c(\beta,\ell,N,p) = 2(\beta\ell(N+1)\Psi^{-1}(p))^2 V$.*

Note that the required number is of the second order $\mathcal{O}((1/\Delta\ell)^2)$, which means one should be particularly careful when comparing two models with close strength. For example, when we set $\beta = 0.35, \ell = 0.7, N = 43$ and $p = 5\%$ we have $c \approx 2.85$ where we use the numerical variance $V = 7.4 \times 10^{-3}$. In this situation, when $\Delta\ell = 0.05$, $N_q \geq 1140$, while when $\Delta\ell = 0.01$, $N_q \geq 28500$ which cannot be easily satisfied.

### 4.3 Correlation between missing and misclassification

In the previous two subsections, we analyze the degradation and inconsistency with independence assumption between missing facts and model predictions. In some conditions, there could be correlation between the missing data and the trained models. Let us give some examples:

- The missing facts in closed-world KG $G_{full}$ follow some non-uniform distribution. For example, in some KGs, the missing facts are more frequently related to some certain entities. The model could be under-trained on these entities because there are more missing facts related to them when training. And when the model is tested, the queries with more missing test answers correspond to the lower predicting capacity.

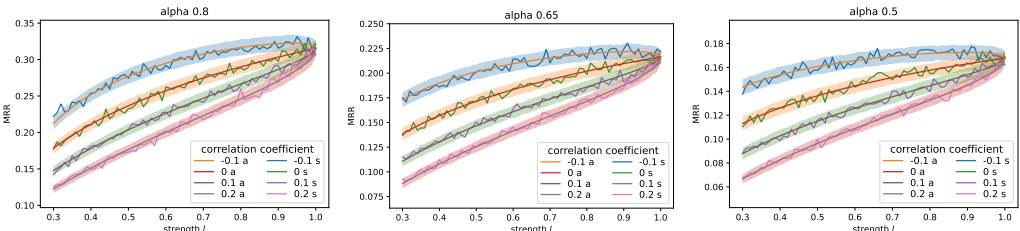

Figure 2: Theoretical approximation (a) and numerical simulation (s) of the expectation with correlation coefficient $\rho$. Details of the figures are the same as in Figure 1.

- The target KG has been preliminarily complemented by some models. In this situation, the missing facts show a negative correlation with the predictive power of this type of models.

Next, we will extend the previous analysis to the situation without the independence assumption. For this goal, we use the correlation efficient to model the correlation between random event $X$: a fact is missing and $Y$: this fact is predicted by the model.

**Definition 4.1** (Correlation between fact missing and model prediction). The correlation coefficient $r$ between two random events $X$ and $Y$ can be defined as follows.

$$\rho = \frac{P(XY) - P(X)P(Y)}{\sqrt{P(X)P(\overline{X})P(Y)P(\overline{Y})}}. \tag{7}$$

With the correlation coefficient $\rho$, the prediction accuracy on missing answers $e \in G_{full} \setminus G_{test}$ and on test answers $e \in G_{test} \setminus G_{train}$ can be calculated as the conditional probability as $P(Y|X) = \ell + \sqrt{\ell(1-\ell)\alpha/\beta} \cdot \rho$ and $P(Y|\overline{X}) = \ell - \sqrt{\ell(1-\ell)\beta/\alpha} \cdot \rho$. We denote them as $\ell_1$ and $\ell_2$ respectively.

Theorem 4.1 can then be generalized as follows:

**Theorem 4.3.** *We have the approximation for MRR*

$$\mathbb{E}(MRR) \approx \frac{\ell_2}{\ell_1} \cdot \frac{\ln(\ell_1) + \ln(\beta) + \ln(N+2) + \gamma}{\beta(N+1)} := \tilde{\mathbb{E}}. \tag{8}$$

The theoretical error analysis is in Appendix A.1. We also evaluate the approximation by numerical simulation and the results are shown in Figure 2. The approximation fits the numerical simulation well. Comparing different models with the same model strength $\ell$ but different correlation coefficient $\rho$, the models with smaller $\rho$ have the higher MRR. The phenomenon is consistent with Corollary 4.5.

**Corollary 4.5** (The derivative w.r.t $\rho$ with correlation). *If we have the inequality $\ell_1 \beta(N+2) \geq \exp(\alpha + \sqrt{\frac{\alpha\beta(1-\ell)}{\ell}} - \gamma)$, then the derivative $\frac{\partial \tilde{\mathbb{E}}(\mathrm{MRR})}{\partial \rho} < 0$.*

The condition in Corollary 4.5 has requirements for lower bound $\ell_1$ and $\rho$. If $\ell_1$ is close to 0, the condition can be violated. This corollary suggests more severe inconsistency. In a reasonable range, the expectation of MRR is monotonically decreasing w.r.t $\rho$. This conclusion suggests that the metric MRR may favor the models with smaller $\rho$ instead of larger $\ell$. Note that the inconsistency is of expectation, which cannot be solved by more test queries.

## 5 Relationship between focus-on-top and degradation

We have pointed out the degradation and inconsistency under the open-world assumption for some most-used ranking-based metrics. Specifically, the derivative of the metrics w.r.t $\ell$ can be too small to reflect the increasing of the true improvement of the model strength (degradation). According to Corollary 4.1, the derivative is related to the expectation of $g(r) = r/f(r)$ where $r$ follows a binomial distribution. We point out the degradation is due to the too small expectation of $g$ relative to the denominator $N$, which is inherently caused by a property of the metrics called *focus-on-top*.

The focus-on-top property means that the metrics are more sensitive to ranking change in top places. For example, MRR changes from 1 to 0.5 when the ranking changes from 1 to 2, but only changes

from 1e-2 to 0.99e-2 when the ranking changes from 100 to 101. This property can simulate the human behavior that people pay more attention to the top answers. However, under the open-world assumption, the focus-on-top property causes negative impacts by making the function $1/f(\boldsymbol{r})$ decrease too fast so that the expectation of $g$ is too small relative to $N$. For example, according to Corollary 4.3, a smaller $K$ means more focus-on-top and smaller derivative.

We can also understand the relationship intuitively. Focusing-on-top means that a few missing answers can have a large impact on the metric, especially when the model performance is already good and the rankings of the rest answers fall into the sensitive range. It is also consistent with our observation that the flatting problem is more severe when the strength $\ell$ increases.

There is a trade-off between focus-on-top and consistency. Therefore, one solution to the degradation and inconsistency is to add in some less focus-on-top metrics as a verification when evaluating, which have a relatively slower descending rate. For example, the $\log$-MRR where $f(r) = \log_2(r+1)$ and $p$-MRR where $f(r) = r^p$, $0 < p < 1$ are less focus-on-top than the standard MRR. If the conclusions of these less focus-on-top metrics are consistent with the MRR or Hits@K, the credibility of the conclusions will be greatly enhanced.

# 6 Experiments on an artificial KG

In this section, we aim to conduct experiments with practical KGC models on a meaningful closed-world KG to further verify our conclusions. The reason we want a **closed-world KG** is for comparing the reported MRR with the true model strength which should be measured on the full test set. To find a closed-world KG, however, it is impractical to resort to existing real-world ones since we have no guarantee that the KG has no missing facts. Therefore, we must resort to some artificial KGs.

For this purpose, we generate an artificial **family tree KG**, which contains 6,004 entities, 23 relations, and 192,532 facts. The relation set contains all common family relations such as parent, child, husband, wife, sister, and brother. The details are included in Appendix A.3. The generated KG is **closed-world** since all the facts can be deduced by a symbolic reasoning tool called DLV system [Leone et al., 2006] (free for academic use). With the closed-world KG, we can simulate the practical open-world setting by artificially controlling the degree of random fact missing, which is measured by density $d = |G_{test}|/|G_{full}|$. The relation between $d$ and $\alpha$ is explained in Appendix A.3.

The generated KG may be representative of a class of real-world KGs with rich rules and simple relations. However, we admit that for Wiki-KGs such as Freebase[Bollacker et al., 2008], there may be a certain interval, but the closed-form solution KG corresponding to the latter is impossible to obtain.

With the closed-world KG, we aim to verify our previous conclusions restated below:

1. There is metric degradation which means the curves of metric increasing become flatter and flatter with the increasing of model strength $\ell$. Further, the degradation may result in inconsistency, where stronger models report lower metric numbers.

2. Considering the correlation between fact missing and model prediction, if the correlation degrees vary among different models, the inconsistency problem may become more severe.

3. The degrees of degradation and inconsistency are related to the focus-on-top property of the metrics. With less focus-on-top metrics, these problems could be relieved.

Our code and data are available at https://github.com/GraphPKU/Open-World-KG. The experiments were run on two clusters with four NVIDIA A40 and six NVIDIA GeForce 3090 GPUs respectively.

## 6.1 Degradation and inconsistency under independence assumption

We train four KGC models with different hyperparameter settings (which results in 18 different models in total) and test them on full test set $G_{full} \setminus G_{train}$ and sparse test set $G_{test} \setminus G_{train}$ respectively. The full test metric can be considered as a measurement of model strength $\ell$ which is what we really want to measure, while the sparse test metric is what we can observe in practice. We plot the sparse-full test curve under different densities in Figure 3, where each curve represents a model whose label is shown in the right legend, and the details of the models are given in Appendix A.4.

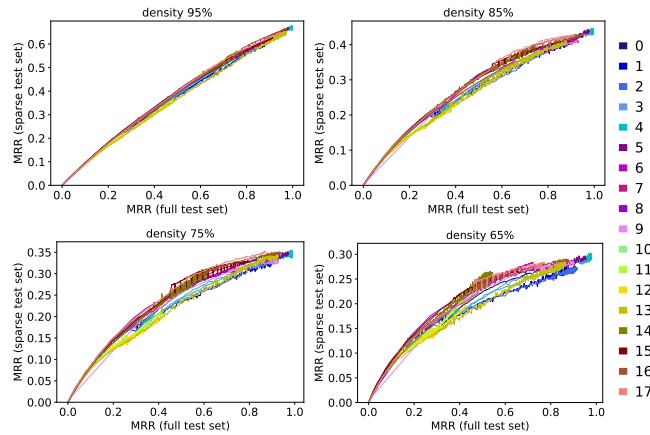
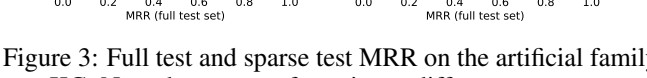
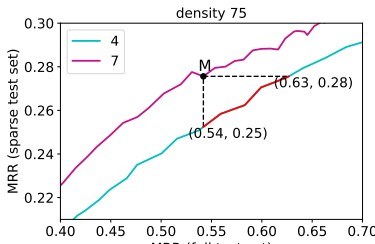

Figure 4: A zoom-in of two curves 4 and 7 under density $d = 0.75$. The checkpoints of model 4 lying on the red segment all have better full test MRR than the checkpoint M of model 7, while reporting worse sparse test MRR under the open-world setting.

Figure 3: Full test and sparse test MRR on the artificial family tree KG. Note the ranges of y-axis are different.

From the figure, we first observe that these curves are indeed shaped like *log* curves. The increasing of the sparse test MRR is slower and slower with the increasing of the full test MRR. Due to the flatting of the curves, the same sparse MRR has a rather broad interval of the corresponding full metric. This phenomenon indicates the degradation of the metric MRR. And as the sparsity increases, the range of the y-axis shrinks (i.e., the curves become flatter), which means the degradation is more severe. Further, these results demonstrate the inconsistency problem of MRR. To illustrate this point more clearly, we zoom in a part of the full figure with two curves as shown in Figure 4. For the model checkpoint corresponding to point M on curve 7, any model checkpoint corresponding to a point on the red segment of curve 4 is **actually stronger than that model**, but reports **a lower sparse MRR**.

## 6.2 Correlation between fact missing and model prediction

In this part, we will simulate the third example we provided in Section 4.3 to check our theory considering the correlation between fact missing and model prediction. Here we use one of the trained ComplEx model (labeled as 16) [Trouillon et al., 2016] to predict on the full test set $G_{full} \setminus G_{train}$ and use its predictions to choose the test set $G_{test} \setminus G_{train}$ and missing facts $G_{full} \setminus G_{test}$. The missing facts are highly correlated with this ComplEx model and therefore could be correlated with other models according to the correlation between different frameworks and model settings. Then we test the other models except for this ComplEx model on the correlated test set. The results of density $d = 75\%$ are shown in Figure 5a and others are shown in Appendix A.5. Though the correlation coefficient is not available for these different models, we indeed observe the gaps between different models become larger than the independent setting, which suggests the inconsistency is more severe.

## 6.3 Less focus-on-top metrics

The next conclusion is that with less focus-on-top metrics, the degradation and inconsistency can be relieved. In Figure 5b, we show the sparse-full curves with some less focus-on-top metrics (log-MRR and $p$-MRR) for the experiments from Section 6.1. Their curves are more close to $y = x$ instead of the *log* function. The flatting is less significant due to the wider range of y-axis. We also observe that the gaps between different models become smaller, which indicates inconsistency is also relieved. Additional results with more metrics, density $d$ and correlation are shown in Appendix A.6.

## 7 Conclusion and future work

In this paper, we study KGC evaluation under the open-world assumption. Theoretically, we model KGC as a positive-negative classification and then deduce an approximation of the expectation of the ranking-based metrics with or without the independence assumption. According to the approximation, we illustrate the degradation and inconsistency of these metrics under the open-world assumption. Furthermore, we point out the focus-on-top property of ranking-based metrics worsen the degradation

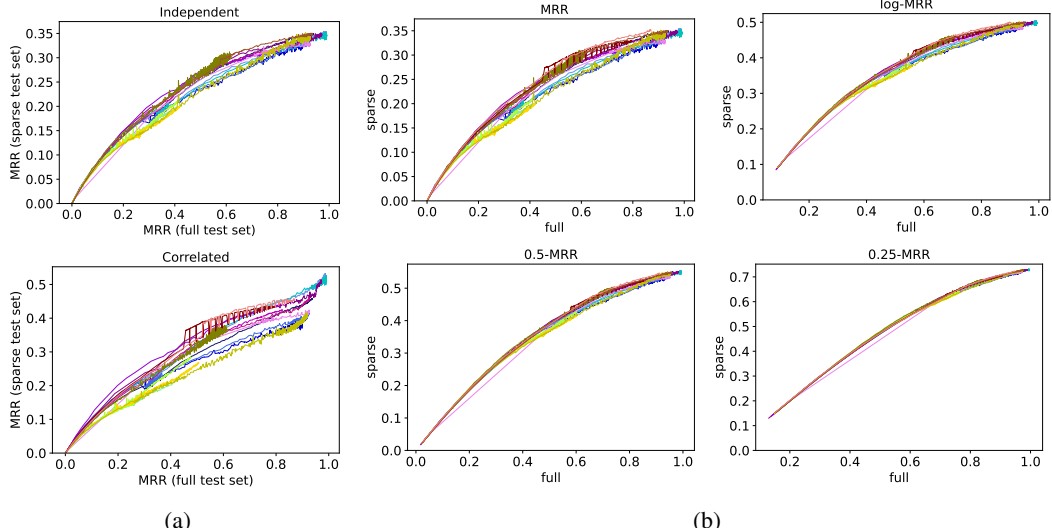

Figure 5: (a) Full test and sparse test MRR on independent (above) and correlated (bottom) family tree KG. (b) MRR, and less focus-on-top metrics. Both are under density $d = 75\%$

and inconsistency. Finally, we generate a closed-world family tree KG and do experiments to verify our theoretical conclusions.

There is still some future work. First, our analysis is based on the positive-negative classification model, which may be too idealistic. In practice, the ranking in positive and negative parts may not be uniformly at random. A more realistic modeling of the KGC task is a direction for our future research. In addition, the correlation between missing facts and prediction could be more complex than our analysis. Finally, we are curious about the possibility to find a more fundamental solution to the open-world problem, which we leave for future work.

## Acknowledgments

Z. Lin was supported by the major key project of PCL (grant no. PCL2021A12), the NSF China (No.s 62276004 and 61731018), and Project 2020BD006 supported by PKU-Baidu Fund. M. Zhang is supported by the NSF China (No. 62276003) and CCF-Baidu Open Fund (NO.2021PP15002000).

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
