# A Appendix

## A.1 Proof

### A.1.1 Lemma 4.1

*Proof.* First, according to the linearity of expectation, we have

$$\mathbb{E}(\mathfrak{M}) = \mathbb{E}_{\boldsymbol{m}} \mathbb{E}_{\boldsymbol{r}} \left( \frac{1}{N-\boldsymbol{m}} \sum_{i=1}^{N-\boldsymbol{m}} \frac{1}{f(\boldsymbol{r}(e_i))} \middle| \boldsymbol{m} \right) = \mathbb{E}_{\boldsymbol{m}} \mathbb{E}_{\boldsymbol{r}} \left( \frac{1}{f(\boldsymbol{r}(e))} \middle| \boldsymbol{m} \right),$$

where $\mathbb{E}(\frac{1}{f(\boldsymbol{r}(e))}|\boldsymbol{m}) = \mathbb{E}(\frac{1}{f(\boldsymbol{r}(e_1))}|\boldsymbol{m}) = \cdots = \mathbb{E}(\frac{1}{f(\boldsymbol{r}(e_{N-\boldsymbol{m}}))}|\boldsymbol{m})$, and here $\{e_1, e_2, \ldots, e_{N-\boldsymbol{m}}\}$ is the test answer set. We denote the final item as $\mathbb{E}(\frac{1}{f(\boldsymbol{r}(e))})$. Then, using the conditional expectation, we have

$$\mathbb{E}\left( \frac{1}{f(\boldsymbol{r}(e))} \right) = P(Y) \mathbb{E}\left( \frac{1}{f(\boldsymbol{r}(e))} \middle| Y \right) + (1 - P(Y)) \mathbb{E}\left( \frac{1}{f(\boldsymbol{r}(e))} \middle| \overline{Y} \right)$$

$$= \ell \, \mathbb{E}\left( \frac{1}{f(\boldsymbol{r}(e))} \middle| Y \right) + (1 - \ell) \mathbb{E}\left( \frac{1}{f(\boldsymbol{r}(e))} \middle| \overline{Y} \right).$$

For the first item, because the ranking of each positive entity is uniformly at random and note the ranking is filtered, we have $P(\boldsymbol{r} = k|Y) = 1/(\boldsymbol{m}+1)$, $\forall k = 1, 2, \ldots, \boldsymbol{m}+1$. Note $\boldsymbol{m}$ is a random variant following the binomial distribution $\mathcal{B}(N, \beta\ell)$, we have

$$\mathbb{E}\left( \frac{1}{f(\boldsymbol{r}(e))} \middle| Y \right) = \mathbb{E}_{\boldsymbol{m}} \left( \frac{1}{\boldsymbol{m}+1} \sum_{k=1}^{\boldsymbol{m}+1} \frac{1}{f(k)} \right)$$

$$= \sum_{m=0}^{N} \binom{N}{m} (\ell\beta)^m (1 - \ell\beta)^{N-m} \frac{1}{m+1} \left( \sum_{k=1}^{m+1} \frac{1}{f(k)} \right)$$

$$= \sum_{k=1}^{N+1} \frac{1}{f(k)} \sum_{m=k-1}^{N} \binom{N}{m} (\ell\beta)^m (1 - \ell\beta)^{N-m} \frac{1}{m+1}$$

$$= \frac{1}{\ell\beta(N+1)} \sum_{k=1}^{N+1} \frac{1}{f(k)} \sum_{m=k-1}^{N} \binom{N+1}{m+1} (\ell\beta)^{m+1} (1 - \ell\beta)^{N-m}$$

$$= \frac{1}{\ell\beta(N+1)} \sum_{k=1}^{N+1} \frac{1}{f(k)} P(\boldsymbol{m} \geq k)$$

$$= \frac{1}{\ell\beta(N+1)} \sum_{k=0}^{N} \frac{1}{f(k+1)} (1 - \hat{\Phi}(k)),$$

where $\hat{\Phi}$ is the cdf of binomial distribution $\mathcal{B}(N+1, \ell\beta)$. To prove this lemma, we only need to prove $0 < (1-\ell) \mathbb{E}(\frac{1}{f(\boldsymbol{r}(e))}|\overline{Y}) \leq (1-\ell) \frac{\ln(N_{entity} - N)}{N_{entity} - N}$. The left side is obvious, and the right side can be proved as follow. Here we use $N_e$ to denote the number of entity instead of $N_{entity}$. Condition on $\boldsymbol{m}$, note that the minimal ranking of negative entities is $\boldsymbol{m}+1$ because there have been $\boldsymbol{m}$ missing answers with higher rankings, and the maximal ranking of them is $N_e - (N - \boldsymbol{m})$ which is the number of entities except for the filtered ones. So we have

$$\mathbb{E}\left( \frac{1}{f(\boldsymbol{r}(e))} \middle| \overline{Y} \right) = \frac{1}{N_e - N} \mathbb{E}_{\boldsymbol{m}} \left( \sum_{k=\boldsymbol{m}+1}^{N_e - N + \boldsymbol{m}} \frac{1}{k} \right) \leq \mathbb{E}_{\boldsymbol{m}} \frac{\ln(N_e - N + \boldsymbol{m}) - \ln(\boldsymbol{m})}{N_e - N} \leq \frac{\ln(N_e - N)}{N_e - N}.$$

The first inequality is because $1/k \leq \ln(k) - \ln(k-1)$. This proves the lemma. $\qquad\square$

### A.1.2 Corollary 4.1

*Proof.* We need the derivative of the cdf of binomial distribution. Assuming $\Phi$ is the cdf of $\mathcal{B}(N, p)$, we have

$$\frac{\mathrm{d}\Phi(K)}{\mathrm{d}p} = \sum_{k=0}^{K} \frac{\mathrm{d}}{\mathrm{d}p} \binom{N}{k} p^k (1-p)^{N-k}$$

$$= \sum_{k=0}^{K} \binom{N}{k} kp^{k-1}(1-p)^{N-k} - \sum_{k=0}^{K} \binom{N}{k}(N-k)p^k(1-p)^{N-k-1}$$

$$= \sum_{k=0}^{K} \binom{N}{k} kp^{k-1}(1-p)^{N-k} - \sum_{k=0}^{K} \binom{N}{k+1}(k+1)p^k(1-p)^{N-k-1}$$

$$= \sum_{k=0}^{K} \binom{N}{k} kp^{k-1}(1-p)^{N-k} - \sum_{k=1}^{K+1} \binom{N}{k} kp^{k-1}(1-p)^{N-k}$$

$$= -\binom{N}{K+1}(K+1)p^K(1-p)^{N-K-1}$$

So for $\hat{\Phi}(k)$ is the cdf of $\mathcal{B}(N+1, \ell\beta)$, we have

$$\frac{\partial \hat{\Phi}(k)}{\partial \ell} = -\beta \binom{N+1}{k+1}(k+1)(\ell\beta)^k(1-\ell\beta)^{N-k}$$

and

$$\frac{\mathrm{d}\hat{\mathbb{E}}}{\mathrm{d}\ell} = \frac{1}{\ell\beta(N+1)} \sum_{k=0}^{N} \frac{k+1}{f(k+1)} \binom{N+1}{k+1}(\ell\beta)^{k+1}(1-\ell\beta)^{N-k}$$

$$= \frac{1}{\ell\beta(N+1)} \sum_{k=1}^{N+1} \frac{k}{f(k)} \binom{N+1}{k}(\ell\beta)^k(1-\ell\beta)^{N+1-k}$$

$$= \frac{1}{\ell\beta(N+1)} \mathbb{E}_{k \sim \mathcal{B}(N+1, \ell\beta)} \, g(k).$$

The final equation is because $g(0) = 0$. □

### A.1.3 Corollary 4.2

*Proof.* For MRR, $g(r) = 1$, $\forall r \in \mathbb{N}_+$ and $g(0) = 0$. Just replace $g$ into Corollary 4.1, we can get this corollary. □

### A.1.4 Corollary 4.3

*Proof.* According to the Corollary 4.1, we have

$$\frac{\mathrm{d}\hat{\mathbb{E}}(\text{Hits@K})}{\mathrm{d}\ell} = \frac{1}{N+1} \sum_{k=1}^{K} k \binom{N+1}{k}(\ell\beta)^{k-1}(1-\ell\beta)^{N+1-k}$$

$$= \sum_{k=1}^{K} \binom{N}{k-1}(\ell\beta)^{k-1}(1-\ell\beta)^{N-(k-1)}$$

$$= \Phi(K-1),$$

where $\Phi$ is the cdf of the binomial distribution $\mathcal{B}(N, \ell\beta)$. □

### A.1.5 Theorem 4.1

*Proof.* Let $\mathbb{E}' = \beta\hat{\mathbb{E}} = \frac{1}{N+1} \sum_{k=0}^{N} \frac{1}{k+1}(1 - \hat{\Phi}(k))$ and $t = \ell\beta$. In the same way in Corollary 4.1, we have

$$\frac{\mathrm{d}\mathbb{E}'}{\mathrm{d}t} = \frac{1}{t(N+1)} \sum_{k=0}^{N} \binom{N+1}{k+1} t^{k+1}(1-t)^{N-k} = \frac{1-(1-t)^{N+1}}{t(N+1)}.$$

For $0 < t_0 < t < 1$, we have

$$\frac{1 - (1 - t_0)^{N+1}}{t(N+1)} \leq \frac{\mathrm{d}\,\mathbb{E}'}{\mathrm{d}t}\bigg|_t \leq \frac{1}{t(N+1)}.$$

Then we integrate them from $t_0$ to 1.

$$-\frac{1 - (1 - t_0)^{N+1}}{N+1} \cdot \ln(t_0) \leq \mathbb{E}'\,|_{t=1} - \mathbb{E}'\,|_{t=t_0} \leq -\frac{1}{N+1} \cdot \ln(t_0).$$

Because $t_0$ is arbitrary, we replace $t_0$ as general $\ell\beta$.

$$-\frac{1 - (1 - \ell\beta)^{N+1}}{N+1} \cdot (\ln(\ell) + \ln(\beta)) \leq \hat{\mathbb{E}}|_{\ell=\beta=1} - \mathbb{E}' \leq -\frac{1}{N+1} \cdot (\ln(\ell) + \ln(\beta)).$$

Note that

$$\tilde{E}|_{\ell=\beta=1} = \hat{E}|_{\ell=\beta=1} = \frac{1}{N+1} \sum_{k=1}^{N+1} \frac{1}{k} = \frac{\ln(N+2) + \gamma - \varepsilon_{N+1}}{N+1}.$$

We denote it as $\mathbb{E}_1$, where $\varepsilon_{N+1} = \ln(N+2) + \gamma - \sum_{k=1}^{N+1} \frac{1}{k}$ is the residual of the sum of harmonic series and $0 < \varepsilon_{N+1} \leq \frac{1}{2(N+1)}$. Then, we have

$$\left(1 - (1 - \ell\beta)^{N+1}\right) \cdot \left(\mathbb{E}_1 + \frac{\varepsilon_{N+1}}{N+1} - \beta\tilde{\mathbb{E}}\right) \leq \mathbb{E}_1 - \mathbb{E}' \leq \left(\mathbb{E}_1 + \frac{\varepsilon_{N+1}}{N+1} - \beta\tilde{\mathbb{E}}\right).$$

For the second inequality, it is equivalent to

$$\hat{\mathbb{E}} - \tilde{\mathbb{E}} \geq -\frac{\varepsilon_{N+1}}{\beta(N+1)} \geq -\frac{1}{2\beta(N+1)^2}.$$

For the first inequality, we have

$$
\begin{aligned}
\hat{\mathbb{E}} - \tilde{\mathbb{E}} &\leq \left(\frac{\mathbb{E}_1}{\beta} - \hat{\mathbb{E}}\right)\left(1 - \frac{1}{(1 - (1 - \ell\beta)^{N+1})}\right) - \frac{\varepsilon_{N+1}}{\beta(N+1)} \\
&\leq \left(\frac{\mathbb{E}_1}{\beta} + \frac{\varepsilon_{N+1}}{\beta(N+1)} - \tilde{\mathbb{E}}\right)\frac{(1 - \ell\beta)^{N+1}}{1 - (1 - \ell\beta)^{N+1}} - \frac{\varepsilon_{N+1}}{\beta(N+1)} \\
&= \frac{(1 - \ell\beta)^{N+1}}{1 - (1 - \ell\beta)^{N+1}} \cdot \frac{\ln(1/(\ell\beta))}{\beta(N+1)} - \frac{\varepsilon_{N+1}}{\beta(N+1)} \\
&\leq \frac{(1 - \ell\beta)^{N+1}}{1 - (1 - \ell\beta)^{N+1}} \cdot \frac{\ln(1/(\ell\beta))}{\beta(N+1)}.
\end{aligned}
$$

Therefore, we have the error bound:

$$|\hat{\mathbb{E}} - \tilde{\mathbb{E}}| \leq \max\left\{\frac{1}{2\beta(N+1)^2}, \frac{(1 - \ell\beta)^{N+1}}{1 - (1 - \ell\beta)^{N+1}} \cdot \frac{\ln(1/(\ell\beta))}{\beta(N+1)}\right\}$$

$\qquad\qquad\qquad\qquad\qquad\qquad\qquad\qquad\qquad\qquad\qquad\qquad\qquad\qquad\qquad\qquad$ □

### A.1.6 Theorem 4.2

*Proof.* Given all the independence assumption, $\mathfrak{M}(\mathcal{M}_2) - \mathfrak{M}(\mathcal{M}_1)$ follows normal distribution $\mathcal{N}(\frac{\ln(1+\frac{\Delta\ell}{\ell})}{\beta(N+1)}, \frac{V(\beta,\ell)+V(\beta,\Delta\ell+\ell)}{N_q})$. So

$$Z = \frac{\mathfrak{M}(\mathcal{M}_2) - \mathfrak{M}(\mathcal{M}_1) - \frac{\ln(1+\frac{\Delta\ell}{\ell})}{\beta(N+1)}}{\sqrt{\frac{V(\beta,\ell)+V(\beta,\Delta\ell+\ell)}{N_q}}} \sim \mathcal{N}(0,1).$$

Then $\mathfrak{M}(\mathcal{M}_2) \leq \mathfrak{M}(\mathcal{M}_1)$ is equivalent to

$$Z \leq \frac{-\frac{\ln(1+\frac{\Delta\ell}{\ell})}{\beta(N+1)}}{\sqrt{\frac{V(\beta,\ell)+V(\beta,\Delta\ell+\ell)}{N_q}}} = -\frac{\sqrt{N_q}\ln(1+\frac{\Delta\ell}{\ell})}{\beta(N+1)\sqrt{V(\beta,\ell)+V(\beta,\ell+\Delta\ell)}}.$$

So the probability is as shown in the theorem. $\qquad\qquad\qquad\qquad\qquad\qquad\qquad$ □

### A.1.7 Corollary 4.4

*Proof.* Just solve $N_q$ from the Theorem 4.2. $\qquad\square$

### A.1.8 Theorem 4.3

*Proof.* We can generalize the Lemma 4.1 as follows.

**Lemma A.1** (Expectation with Correlation). *Under the same assumptions as the lemma 4.1 and the correlation coefficient is $\rho$, the expectation of the metric $\mathfrak{M}$:*

$$\mathbb{E}(\mathfrak{M}) = \frac{\ell_2}{\ell_1} \cdot \frac{1}{\beta(N+1)} \sum_{k=0}^{N} \frac{1}{f(k+1)} \left(1 - \tilde{\Phi}(k)\right) + \delta', \tag{9}$$

*where $\tilde{\Phi}$ is the cdf of binomial distribution $\mathcal{B}(N+1, \ell_1\beta)$ and $0 \le \delta' \le (1 - \ell_2)\frac{\ln(N_{entity} - N)}{N_{entity} - N}$.*

The proof of the lemma is similar to what we have shown in A.1.1. Given the lemma, the $\hat{E}$ can be similarly expressed as $\frac{\ell_2}{\ell_1} \cdot \frac{1}{\beta(N+1)} \sum_{k=0}^{N} \frac{1}{f(k+1)}(1 - \tilde{\Phi}(k))$.

In the similar way in A.1.5, let $\mathbb{E}' = \ell_1\beta\hat{\mathbb{E}}$ and $t = \ell_1\beta$ we have

$$\left(1 - (1 - \ell_1\beta)^{N+1}\right) \cdot \left(\mathbb{E}_1 + \frac{\ell_2\varepsilon_{(N+1)}}{N+1} - \ell_1\beta\tilde{\mathbb{E}}\right) \le \mathbb{E}_1 - \mathbb{E}' \le \left(\mathbb{E}_1 + \frac{\ell_2\varepsilon_{(N+1)}}{N+1} - \ell_1\beta\tilde{\mathbb{E}}\right).$$

where $\mathbb{E}_1 = \frac{\ell_2(\ln(N+2) + \gamma - \varepsilon_{N+1})}{N+1} = \tilde{\mathbb{E}}|_{\ell_1 = \beta = 1} = \hat{\mathbb{E}}|_{\ell_1 = \beta = 1}$. Also using the same technique, the error bound is

$$\hat{\mathbb{E}} - \tilde{\mathbb{E}} \ge -\frac{\ell_2}{2\ell_1\beta(N+1)^2}$$

and

$$\hat{\mathbb{E}} - \tilde{\mathbb{E}} \le \left(\frac{\mathbb{E}_1}{\ell_1\beta} - \hat{\mathbb{E}}\right)\left(1 - \frac{1}{(1 - (1 - \ell_1\beta)^{N+1})}\right)$$

$$\le \left(\frac{\mathbb{E}_1 + \ell_2\varepsilon_{N+1}}{\ell_1\beta} - \tilde{\mathbb{E}}\right)\frac{(1 - \ell_1\beta)^{N+1}}{1 - (1 - \ell_1\beta)^{N+1}}$$

$$= \frac{(1 - \ell_1\beta)^{N+1}}{1 - (1 - \ell_1\beta)^{N+1}} \cdot \frac{\ell_2\ln(1/(\ell_1\beta))}{\ell_1\beta(N+1)}.$$

The error bound is that

$$|\hat{\mathbb{E}} - \tilde{\mathbb{E}}| \le \max\left\{\frac{\ell_2}{2\ell_1\beta(N+1)^2}, \frac{(1 - \ell_1\beta)^{N+1}}{1 - (1 - \ell_1\beta)^{N+1}} \cdot \frac{\ell_2\ln(1/(\ell_1\beta))}{\ell_1\beta(N+1)}\right\}$$

$\qquad\square$

### A.1.9 Corollary 4.5

*Proof.* Let $\beta(N+1) = c$ and $\ln\beta + \ln(N+2) + \gamma = d$, we have

$$\frac{\partial\tilde{\mathbb{E}}}{\partial\ell_1} = \frac{\ell_2(1 - \ln\ell_1 - d)}{c\ell_1^2}, \quad \frac{\partial}{\partial\ell_2} = \frac{\ln(\ell_1) + d}{c\ell_1},$$

and

$$\frac{\partial\ell_1}{\partial\rho} = \sqrt{\frac{\alpha\ell(1 - \ell)}{\beta}}, \quad \frac{\partial\ell_2}{\partial\rho} = -\sqrt{\frac{\beta\ell(1 - \ell)}{\alpha}}.$$

So the derivative w.r.t $\rho$

$$\frac{\partial\mathbb{E}}{\partial\rho} = \frac{\sqrt{\ell(1 - \ell)}}{c\ell_1^2}\left(\ell_2(1 - \ln\ell_1 - d)\sqrt{\frac{\alpha}{\beta}} - \ell_1(\ln(\ell_1) + d)\sqrt{\frac{\beta}{\alpha}}\right)$$

$$= \frac{\sqrt{\ell(1 - \ell)}}{c\ell_1^2}\sqrt{\frac{\alpha}{\beta}}\left(\ell_2 - (\ln(\ell_1) + d)\frac{\ell}{\alpha}\right)$$

Note that $\alpha + \beta = 1$. Because of the conditions $\ell_1 \beta(N+2) \geq \exp(\alpha + \sqrt{\frac{\alpha\beta(1-\ell)}{\ell}} - \gamma)$, we have

$$\ln(\ell_1) + d \geq \alpha + \sqrt{\frac{\alpha\beta(1-\ell)}{\ell}}$$

and then

$$\frac{\ell}{\alpha}\left(\ln(\ell_1) + d)\right) \geq \ell + \sqrt{\frac{\beta(1-\ell)\ell}{\alpha}} > \ell - \sqrt{\frac{\beta(1-\ell)\ell}{\alpha}}\rho = \ell_2.$$

Combining this inequality with the derivative expression, we have $\frac{\partial \mathbb{E}}{\partial \rho} < 0$. $\qquad\square$

## A.2 Details of the simulation

In the Figures 1 and 2, we choose $N$ as $43 = 14505 \times 30\% \times 1\%$, where we assume $N_{entity} = 14505$ as the same as FB15k-237, the total answers accounts for one percent of all entities and the test set accounts for thirty percent of the total answers. For each $\ell$ and $\alpha$ we repeat the simulation with 500 times to calculate the average MRR and the standard derivation.

## A.3 Details of artificial family tree KG

Our codes are modified from [Hohenecker and Lukasiewicz, 2020] (BSD license) to generate the KG. Firstly, it generates all the parent-child relations and then deduced other relations by a symbolic reasoning systems called DLV system [Leone et al., 2006]. We generate 20 family trees then merge them into a whole. Each family tree has three layer depth and 300 entities, with maximal branching width 20 at each internal node. The final artificial KG has 6,004 entities, 23 relations and 192,532 facts. The relations are listed as follow:

- parentOf
- sisterOf
- brotherOf
- siblingOf
- motherOf
- fatherOf

- wifeOf
- husbandOf
- grandmotherOf
- grandfatherOf
- auntOf
- uncleOf

- girlCousinOf
- boyCousinOf
- cousinOf
- daughterOf
- sonOf
- childOf

- granddaughterOf
- grandsonOf
- grandchildOf
- nieceOf
- nephewOf

Note that $G_{full} \supseteq G_{test} \supseteq G_{train}$. We use density $d$ to denote the ratio $|G_{test}|/|G_{full}|$ and then in the open-world KG $G_{test}$ we split the training set and test set with ratio $\eta = |G_{train}|/|G_{test}|$. For each facts, it is a missing fact with probability $1-d$, a test fact with probability $d(1-\eta)$ and a training fact with probability $d\eta$. We set $\eta = 0.7$ and $d = 95\%, 85\%, 75\%, 65\%$ which corresponds to $\alpha = \frac{|G_{test} \setminus G_{train}|}{|G_{full} \setminus G_{train}|} = \frac{d(1-\eta)}{1-d\eta} = 85\%, 63\%, 47\%, 35\%$.

As the same as [Ren et al., 2020, Ren and Leskovec, 2020], we organize the test by queries which means we firstly randomly sample the test queries $r(e_h, ?)$ and then search answers $e \in G_{full} \setminus G_{test}$ as missing answers and $e \in G_{test} \setminus G_{train}$ as test answers. In order to simulate the real situation of common KGs, we filter out the queries with less than 10 answers in the closed-world graph $G_{full}$. Finally, for each sparsity $d$ we choose 500 test queries. For training, we use all facts in $G_{train}$.

## A.4 Details of models

Different models are trained on artificial family tree KG. During training, we test them on full test set and sparse test set to plot the sparse-full curve to show the inconsistency. We choose different framework, including RotatE, pRotatE [Sun et al., 2019], ComplEx [Trouillon et al., 2016] and BetaE [Ren and Leskovec, 2020]. We also test Q2B [Ren et al., 2020] and TransE [Bordes et al., 2013] models, both of which cannot fit the KG well. For each framework, we use several setting, where their label in Figure 3 and the hyper-parameters of the models are shown in Table 2. Here, We have filtered some models which maximal strength $\ell < 0.1$.

## A.5 Experiments with correlation

Here we show more results of the experiments on the correlated family tree KG in Figure 6.

Table 2: Detail of the models trained on family tree KG.

| label | model | dimension | gamma | step | batchsize | negative sampling |
|---|---|---|---|---|---|---|
| 0 | RotatE | $100^4$ | 24 | 100000 | 1024 | 128 |
| 1 | RotatE | 500 | 12 | 100000 | 256 | 128 |
| 2 | RotatE | 500 | 12 | 100000 | 1024 | 512 |
| 3 | RotatE | 500 | 24 | 100000 | 1024 | 128 |
| 4 | RotatE | 1000 | 24 | 100000 | 1024 | 128 |
| 5 | pRotatE | 1000 | 24 | 12000 | 1024 | 128 |
| 6 | pRotatE | 250 | 24 | 12000 | 1024 | 128 |
| 7 | pRotatE | 500 | 24 | 12000 | 1024 | 128 |
| 8 | pRotatE | 500 | 24 | 12000 | 128 | 512 |
| 9 | pRotatE | 500 | 6 | 12000 | 1024 | 128 |
| 10 | BetaE | 1000 | 60 | 400000 | 1024 | 128 |
| 11 | BetaE | 500 | 240 | 400000 | 1024 | 128 |
| 12 | BetaE | 500 | 60 | 400000 | 1024 | 128 |
| 13 | BetaE | 500 | 15 | 400000 | 1024 | 128 |
| 14 | BetaE | 100 | 60 | 400000 | 1024 | 128 |
| 15 | ComplEx | 1000 | 500 | 100000 | 1024 | 128 |
| 16 | ComplEx | 1000 | 200 | 100000 | 512 | 256 |
| 17 | ComplEx | 2000 | 500 | 100000 | 1024 | 128 |

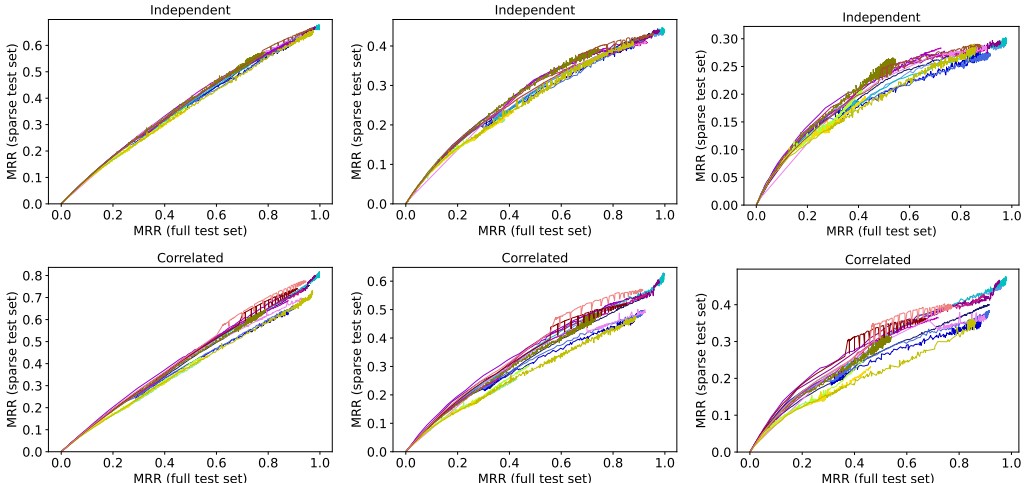

Figure 6: Full test and sparse test MRR on independent (above) and correlated (bottom) family tree KG. Density $d = 95\%, 85\%, 65\%$ from left to right.

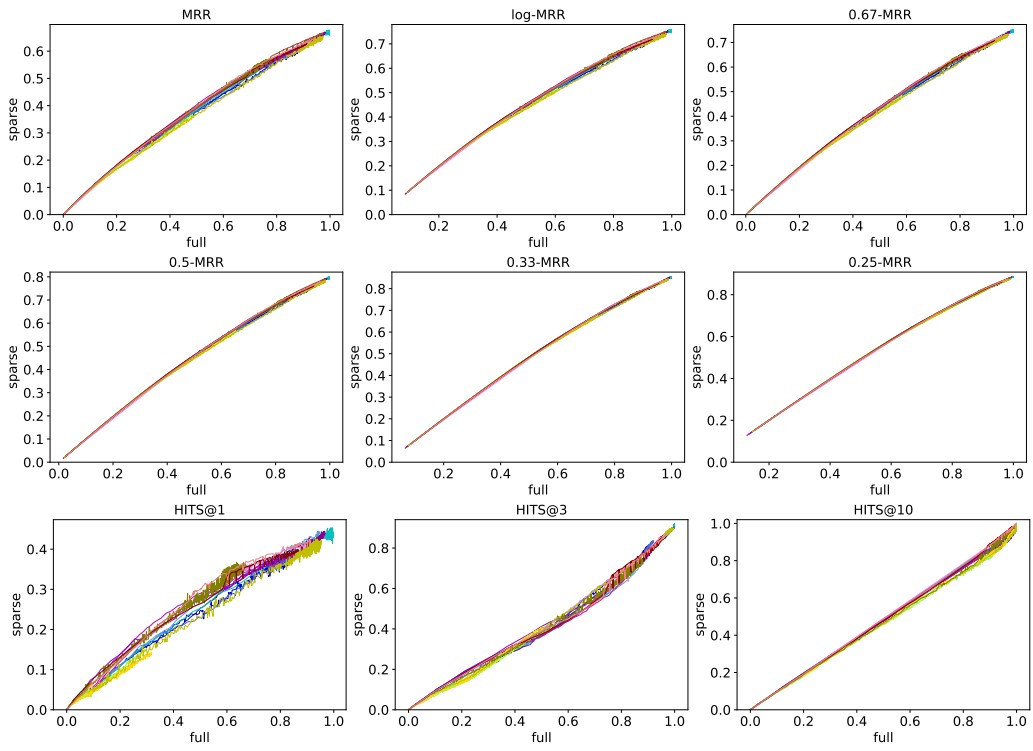

Figure 7: $d = 95\%$ independent

## A.6 Family tree experiments with MRR, Hits@K and more less focus-on-top metrics

The results for other density $d$ and more metrics in the independent situation are shown in Figures 7-10. And the results in the correlated situation are shown in Figures 11-14.

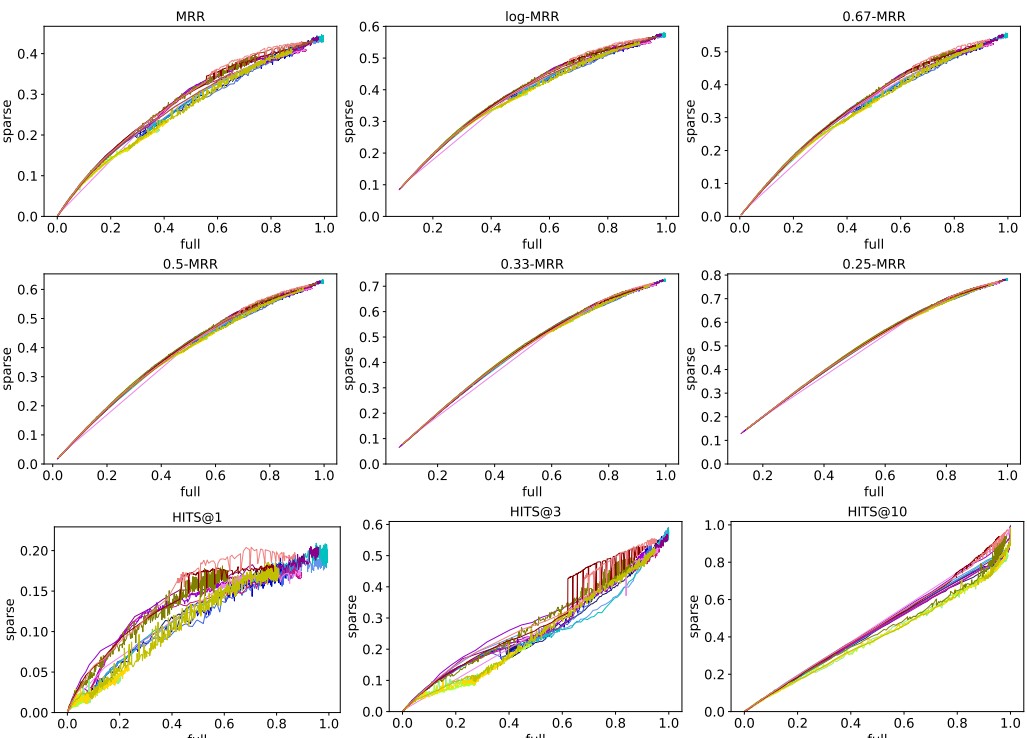

Figure 8: $d = 85\%$ independent

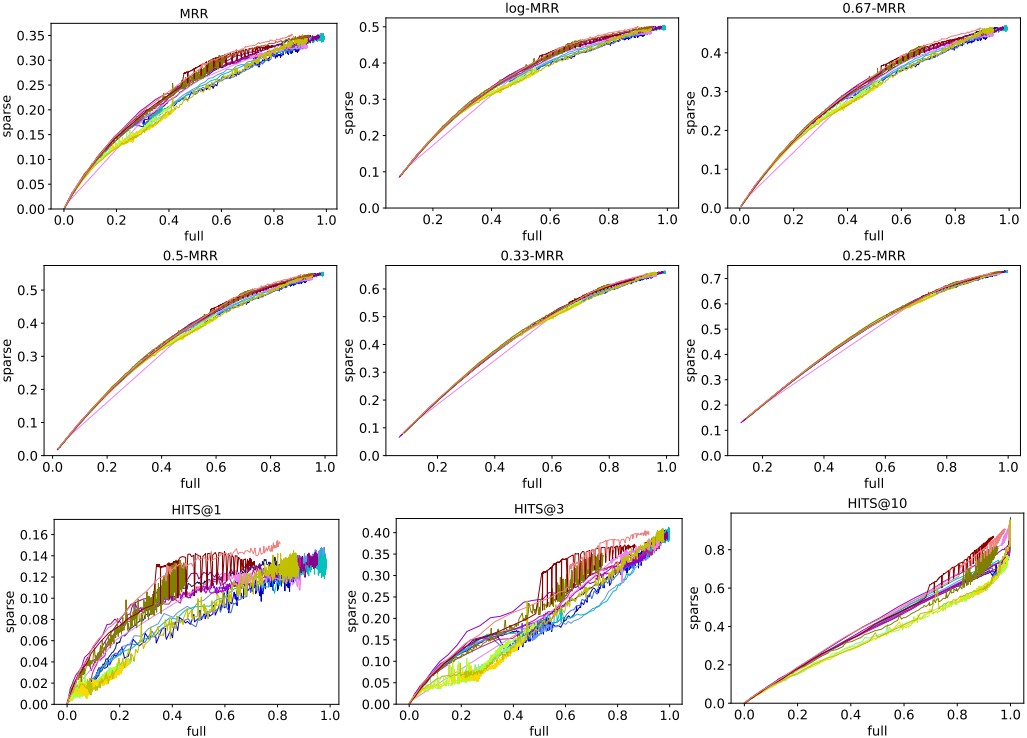

Figure 9: $d = 75\%$ independent

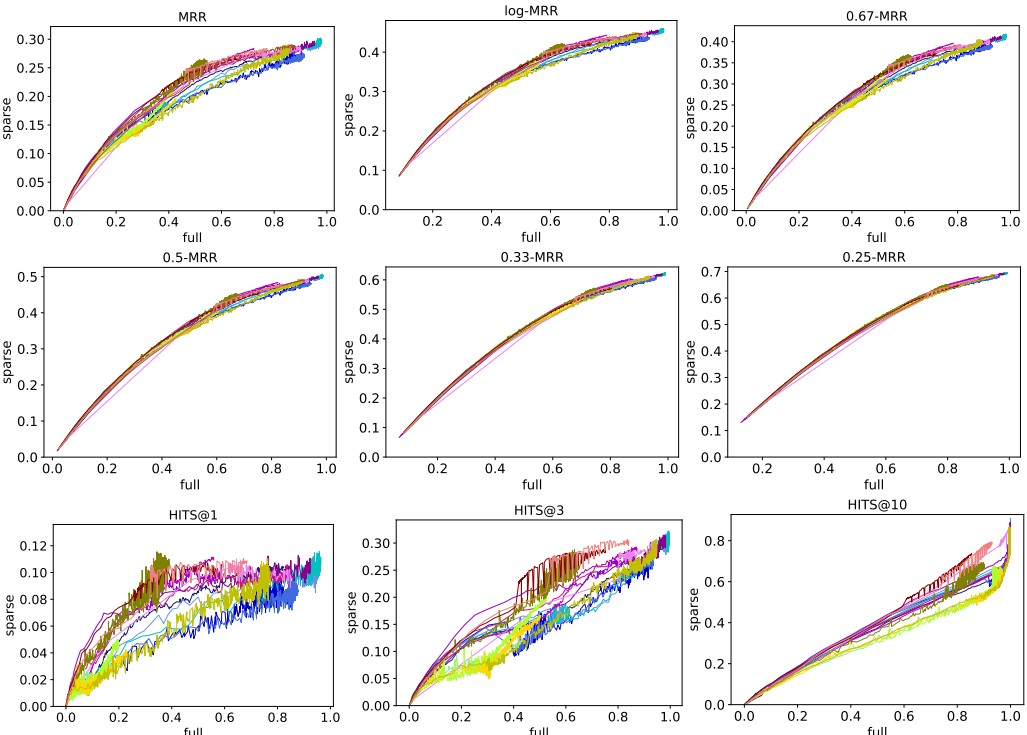

Figure 10: $d = 65\%$ independent

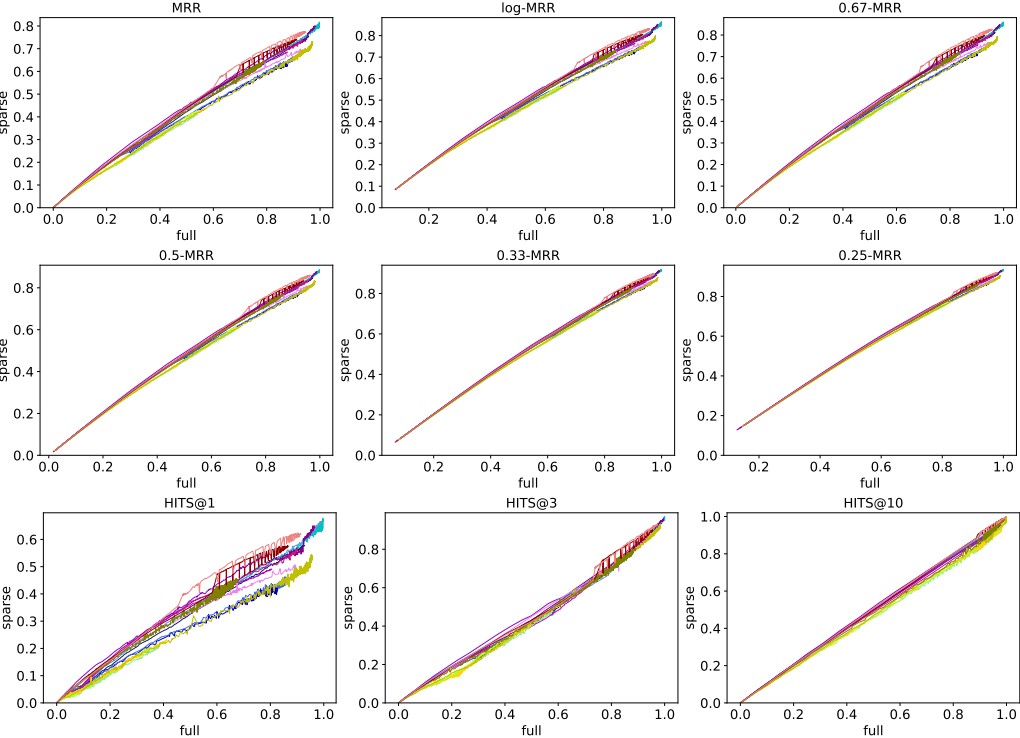

Figure 11: $d = 95\%$ correlated

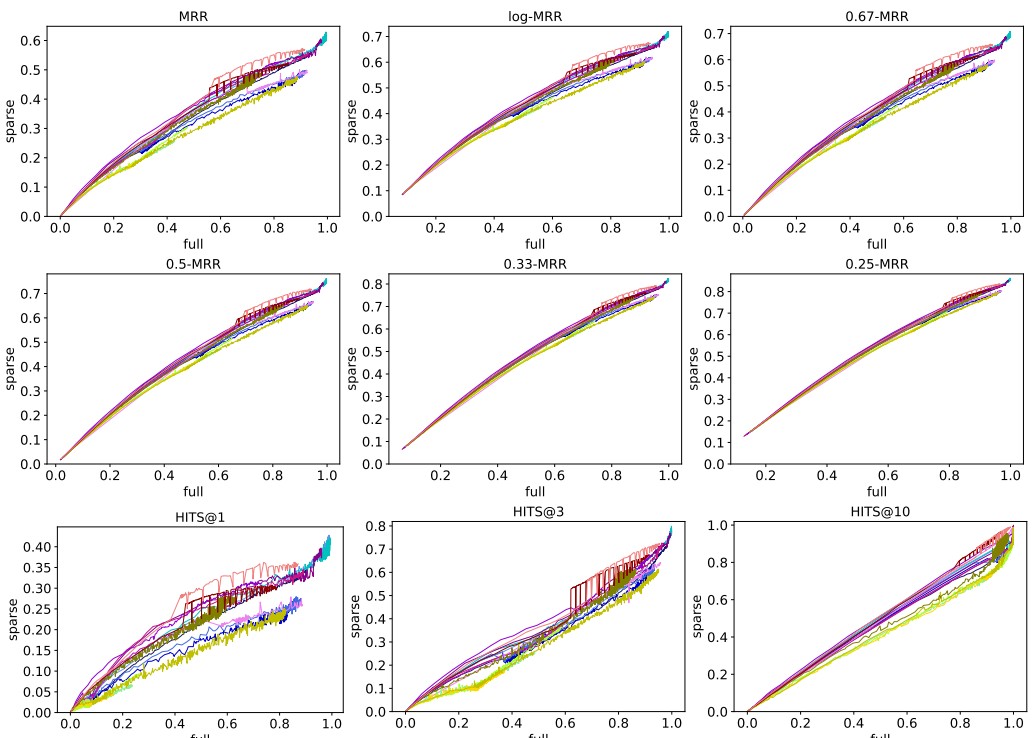

Figure 12: $d = 85\%$ correlated

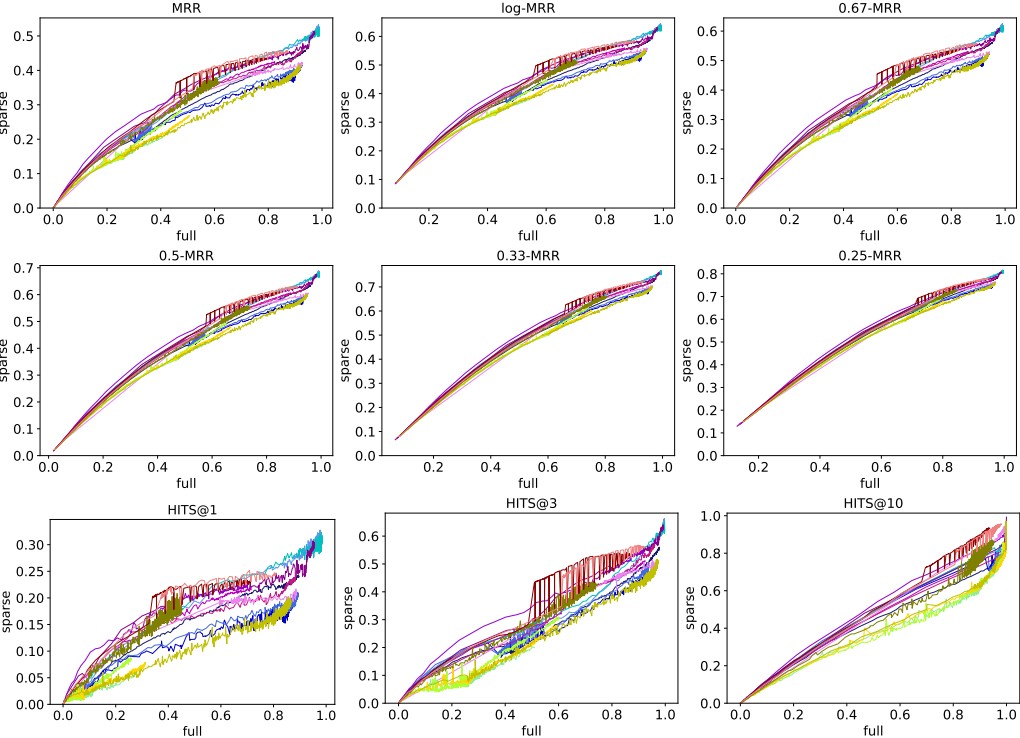

Figure 13: $d = 75\%$ correlated

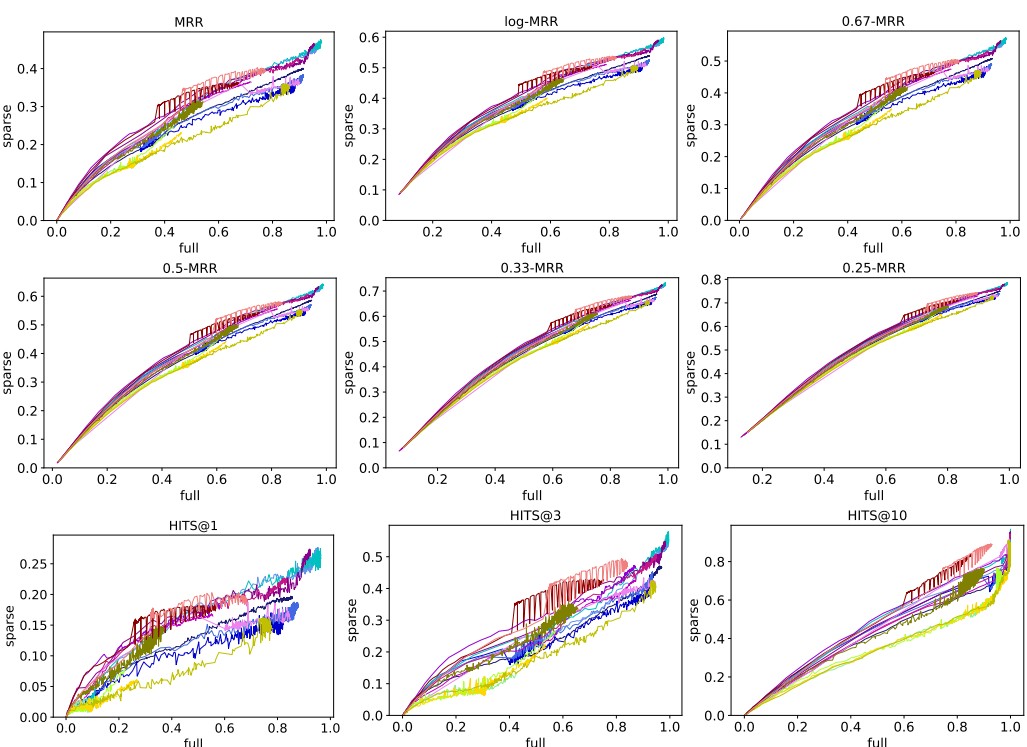

Figure 14: $d = 65\%$ correlated