# OpenReview forum: "Rethinking Knowledge Graph Evaluation Under the Open-World Assumption"
_NeurIPS.cc/2022/Conference — NeurIPS 2022 Accept_

### Official Review · Reviewer_Lj1a · 2022-07-11

**Rating:** 7
**Confidence:** 4
**Soundness:** 3 good
**Presentation:** 3 good
**Contribution:** 3 good

**Summary:**

This paper studies the evaluation of KG completion under the open-world assumption. Its main contribution is to give a theoretical analysis of common evaluation metrics on the open-world knowledge graph and point out the shortcomings of focus-on-top metrics. In addition, experiments on a closed-world dataset validate the obtained conclusions.

**Questions:**

If necessary, the authors can choose to discuss the proposed issues in the above *Strengths And Weaknesses* section to help me make a better assessment of this paper.

**Limitations:**

I would like this paper to be accepted, but there are several limitations of this article that need to be noted. This paper only analyzes several existing evaluation metrics, and does not propose a better solution. In addition, due to its rich set of high-quality rules, the relational pattern of Family Tree KG is much simpler than that of a large-scale knowledge graph, so some basic models can also get high scores. Experimental analysis on such toy KGs may be biased, although I understand that full labeling of large-scale knowledge graphs is impossible.

**Strengths And Weaknesses:**

Pros：

- The paper is well written and the theoretical analysis is easy to understand.

- A fundamental problem of KGC is analyzed, which is beneficial for the development of the field.

Cons:

- There is a work that investigates a similar problem (https://arxiv.org/abs/2108.01387, although its main contribution is not theoretical analysis), but a related discussion is missing.

- Lack of discussion of specific relations. For example, 1-to-1 relations do not have the problem of degradation and inconsistency. However, some 1-to-n relations, especially when n is large, may have this problem.

---

> ### Author Response · Authors · 2022-08-02
> **Author Response**
>
> Thank you for your insightful comments. We address them as follows.
>
> > There is a work that investigates a similar problem (https://arxiv.org/abs/2108.01387, although its main contribution is not theoretical analysis), but a related discussion is missing.
> >
>
> Thanks for your and other reviewers’ reminder. We will add this paragraph to the related work.
>
> - Some recent works have noticed the gap between the actual open-world situation and the closed-world assumption. Cao et al. [2021] point out that the closed-world assumption leads to a trivial evaluation on the triple classification task. They offer their manually-labeled positive-negative-unknown ternary triple classification datasets following the open-world assumption and point out the lack of capacity for current models to distinguish unknown from negative. However, the unknown part in the dataset is only on the triple classification task, while we focus on the link prediction task here. Additionally, Das et al. [2020] analyze the open-world setting as an evolving world that continuously adds new entities and facts to KGs. Under this setting, their work focuses on the inductive or case learning capacity, i.e., the capacity of models to generalize on unobserved entities. Here, we aim to analyze the possible inconsistent comparison in evaluation with missing facts instead of a specific framework with a larger inductive capacity.
>
> > Lack of discussion of specific relations. For example, 1-to-1 relations do not have the problem of degradation and inconsistency. However, some 1-to-n relations, especially when n is large, may have this problem.
> >
>
> Thanks a lot for pointing this out. As you said, for 1-to-1 relations, the problem does not exist. More precisely, for 1-to-1 relations in test sets, because we have observed one answer, we can guarantee the query has been ‘closed’ since we know there is exactly one answer to the query.  In this scenario, there is no risk of ranking other missing facts above the test answer. However, exact 1-to-1 relations are rare in KGs and are hard to ensure in advance. For example, only 5.8% of relations within 237 relations in FB15k-237 are 1-to-1 according to [(Akrami, Farahnaz, et al, 2020)](https://arxiv.org/abs/2003.08001). Therefore, we did not specifically distinguish 1-to-1 relations from others and followed the setting of Query2box ([https://arxiv.org/abs/2002.05969](https://arxiv.org/abs/2002.05969)). We will make this more clear in the revision.
>
> > Due to its rich set of high-quality rules, the relational pattern of Family Tree KG is much simpler than that of a large-scale knowledge graph, so some basic models can also get high scores. Experimental analysis on such toy KGs may be biased, although I understand that full labeling of large-scale knowledge graphs is impossible.
> >
>
> Thanks for the comment. Indeed, we use the rich-rule KG, Family Tree, to meet the closed-world assumption. This Family Tree KG has 23 relations, close to those of Kinship (26 relations) and WN18 (18 relations). Although not perfect, it may be still representative of a class of real-world KGs with rich rules and simple relations, but cannot represent those general wiki KGs such as FreeBase. We will mark this limitation in our revision.

---

### Official Review · Reviewer_c8HZ · 2022-07-12

**Rating:** 8
**Confidence:** 4
**Soundness:** 4 excellent
**Presentation:** 3 good
**Contribution:** 3 good

**Summary:**

This paper revisits the rank based metrics used for evaluating knowledge base completion (KBC).

Large KBs, by definition, are incomplete. Models for KBC are evaluated based on their ranking of all entities given a query entity (head entity) and a query relation with the model being penalized for ranking entities that are not present in the held-out test set higher above.

However, the paradox lies in the fact that certain answers for which models are penalized are actually correct, since that fact might not just be present in the KB (open-world setting)

This paper shows that in the open-world setting, the metrics does not capture the true model improvement as the strength of the model increases. Specifically, as the model strength increases, the metrics show a logarithmic trend, i.e. it might be too flat to capture improvements of stronger models (gets worse as sparsity increases) (metric degradation problem)

Also, more importantly, the metrics might favor a model which would have lower performance than another model if evaluated on a KG with no additional missing edges other than the query (metric inconsistency problem).

Lastly, the paper proposes two metrics which focus less on “entities at top” and shows that they can reduce the degaradation and inconsistency problem.

**Questions:**

The MRR in fig 1 looks surprisingly low. For example, for the most powerful model (strength=1), and with high density, the best MRR only reaches around 0.3? But in practice for KGs such as WN18RR, the MRR goes over 0.5. How do you explain the discrepancy?

In Line 13 of abstract, shouldnt it say “with many missing facts”, instead of “with not many missing facts”?

There are certain papers that model open-world setting as a continuously increasing KB where unknown entities and facts are added. Worth citing them for completion - e.g. Probabilistic CBR for Open-world KBC.

**Limitations:**

Yes, the authors have addressed the limitations of their work.

**Strengths And Weaknesses:**

Strengths:

Originality:

1. This paper studies in detail the efficacy of the ranking of evaluation of KBC models in open-world setting. The analysis is well done and highly novel. The paper shows that the evaluations are not perfect and also proposes couple of metrics which can fix the problems. These are important contributions.

Quality:

2. The paper is high quality work in general. All the claims are justified by both theory, simulation and experiments

Clarity:

3. Very clearly written. The paper was very easy to follow.

Significance:

4. The paper makes fundamental contributions in understanding of evaluation metrics and I think the contributions are significant

---

> ### Author Response · Authors · 2022-08-02
> **Author Response**
>
> Thank you for your constructive comments. We address them as follows.
>
> > Question: The MRR in fig 1 looks surprisingly low. For example, for the most powerful model (strength=1), and with high density, the best MRR only reaches around 0.3? But in practice for KGs such as WN18RR, the MRR goes over 0.5. How do you explain the discrepancy?
> >
>
> According to Equation (5) of our paper, many factors can influence the expected MRR, such as the answer number and the density of datasets. Take the density as an example. In our simulation, we set the density manually to at most 0.8, which could be higher in real KGs (but not measurable). A higher density means that this KG is closer to the closed world, so the MRR will also be higher. For example, the mentioned WN18RR (WordNet) contains relationships between English words and is closer to a closed world than some more general KGs such as FreeBase (which contains entities like people, places, movies, songs that tend to be incomplete). Thus, we indeed see higher MRR numbers reported for WN18RR than for FB15k-237.
>
> > In Line 13 of abstract, shouldnt it say “with many missing facts”, instead of “with not many missing facts”?
> >
>
> We want to express that *'The logarithmic degradation can be **already** observed even with not many missing facts (e.g., about 10% are missing.)'*. With many missing facts, the logarithmic degradation also certainly exists and will be more severe.
>
> > There are certain papers that model open-world setting as a continuously increasing KB where unknown entities and facts are added. Worth citing them for completion - e.g. Probabilistic CBR for Open-world KBC.
> >
>
> Thanks for your and other reviewers’ reminder. We will add this paragraph to the related work.
>
> - Some recent works have noticed the gap between the actual open-world situation and the closed-world assumption. Cao et al. [2021] point out that the closed-world assumption leads to a trivial evaluation on the triple classification task. They offer their manually-labeled positive-negative-unknown ternary triple classification datasets following the open-world assumption and point out the lack of capacity for current models to distinguish unknown from negative. However, the unknown part in the dataset is only on the triple classification task, while we focus on the link prediction task here. Additionally, Das et al. [2020] analyze the open-world setting as an evolving world that continuously adds new entities and facts to KGs. Under this setting, their work focuses on the inductive or case learning capacity, i.e., the capacity of models to generalize on unobserved entities. Here, we aim to analyze the possible inconsistent comparison in evaluation with missing facts instead of a specific framework with a larger inductive capacity.

---

### Official Review · Reviewer_LAYz · 2022-07-12

**Rating:** 4
**Confidence:** 3
**Soundness:** 2 fair
**Presentation:** 3 good
**Contribution:** 2 fair

**Summary:**

The paper investigated the KGC evaluation framework under the open-world assumption. This is a case where unknown triplets are considered to include many missing facts not included in the training or test sets. The paper consider KGC as a positive-negative classification and deduce an approximation of the expectation of the ranking-based metrics with or without the independence assumption.

The currently used metrics such as mean reciprocal rank (MRR) and Hits@K, show an unexpected behavior under the open-world assumption.
Authors point out the focus-on-top property of ranking-based metrics worsen the degradation and inconsistency.
Considering the variance, the paper shows that the degradation in the reported numbers result in incorrect comparisons between different models, where stronger models may have lower metric numbers.


**Questions:**

How to assume freshness, authority, accuracy when adding/detecting a triple when dealing with the open-world?
How to detect the accuracy of a triplet when dealing with open domain?
How to deal with bias, etc.?



**Limitations:**

Please refer to comments above

**Strengths And Weaknesses:**

The paper discusses a very important research topic of big interest to the community. The comparison with current approach showd improvements point toward true north, which is nice to see. The paper consider KGC as a positive-negative classification and deduce an approximation of the expectation of the ranking-based metrics with or without the independence assumption.

Opportunities:
The proposed evaluation is still not convincing yet, even though we see improvements to current techniques. The assumption of positive-negative classification model doesn’t seem ideal. Other assumptions doesn’t seems matching a real world problem. This include the ranking in positive and negative parts that is considered random uniformly.
How to assume freshness, authority, accuracy when adding/detecting a triple when dealing with the open-world.

---

> ### Author Response · Authors · 2022-08-02
> **Author Response 1/2**
>
> Thank you for your insightful comments. We address them as follows.
>
> > The proposed evaluation is still not convincing yet, even though we see improvements to current techniques.
> >
>
> Our main aim is to theoretically analyze the behavior of the rank-based metrics under the open-world assumption. As one of our main conclusion, we point out that the fundamental cause of the degradation is the ‘focus-on-top’ property, which used to be considered a desired property. Instead of proposing some new evaluation, we want to alert researchers in this field to the possible side effects of the ‘focus-on-top’ property. At the same time, we also point out some more robust metrics (e.g., 0.5-MRR and 0.25-MRR in Section 6.3 and Figure 5(b)).
>
> > The assumption of the positive-negative classification model doesn’t seem ideal. Other assumptions doesn’t seems matching a real world problem. This include the ranking in positive and negative parts that is considered random uniformly.
> >
>
> We indeed make some assumptions to facilitate our analysis. Without these assumptions, it is hard to even define some basic concepts, i.e., the actual length of models or the distribution of the missing facts. For example, without a mathematical model of the distribution of missing facts, it is impossible to analyze the open-world problems with probabilistic tools. Here, we choose the binary classification and uniform-tie-breaking assumptions as a simple and uninformative prior. We admit that our assumptions may simplify the actual situation, but below we show that our binary classification and uniform-tie-breaking assumptions are highly reasonable.
>
> On one hand, in the link prediction task, what we **really** want is the answer set of the given query $r(e, ?)$, instead of the ranking of the scores of all entities. The positive-negative classification is actually the **requirement of the task**. Previous works use ranking-based metrics instead of the more reasonable precision/recall because the threshold to divide the positive and negative cannot be readily determined. Factually, for an ideal well-trained model or an **oracle** model (that gives all correct answers a probability 1 and incorrect answers a probability 0), the ranking-based evaluation will be reduced to the binary classification which uniformly breaks ties. On the other hand, even though previous works adopt ranking-based metrics **for evaluation**, modern methods (e.g., [RotatE](https://arxiv.org/abs/1902.10197), [HousE](https://arxiv.org/pdf/2202.07919), and [Query2box](https://arxiv.org/abs/2002.05969) )mostly still model KGC as a **classification task** by **using binary cross-entropy loss** with negative sampling. After all, the ground truth only reveals what are positive answers to a query without providing a ranking for them. All the positive answers to a query are essentially indistinguishable to a model. Thus the uniform-tie-breaking assumption is also reasonable. The final preference output by a model comes from model bias or training randomness.
>
> To further verify whether our assumption is reasonable, we show the score distribution of a real model. We trained the ComplEx model on the Family dataset and fb15k-237 dataset and illustrate the average distribution of output scores on the test set in the figures [Fig1](https://i.imgur.com/w4BSVAB.jpg) and [Fig2](https://i.imgur.com/WLPwNeC.jpg). We can observe the obvious threshold between the answer part and the not-answer part on both artificial and real datasets, suggesting the behavior of the models is just like a binary classification.
>
> Finally, our experiments on the Family dataset do not need those assumptions, and the experimental results are consistent with our theory, showing the logarithm degradation and inconsistency (Figures 3 and 5(a)). This also suggests that our assumptions are reasonable and consistent with the actual situation.

---

> > ### Author Response · Authors · 2022-08-02
> > **Author Response 2/2**
> >
> > > How to assume freshness, authority, accuracy when adding/detecting a triple when dealing with the open-world.
> > >
> >
> > We believe this is an open-ended question for general KG completion tasks. In fact, every real-world KG to complement is open-world (otherwise there is no need to perform the completion since all knowledge is known in a closed-world KG). The freshness, authority and accuracy thus relate to how to deal with time (old knowledge is replaced by new knowledge), how to identify authoritative sources and facts, and how to generally improve the KGC quality. Our theoretical analysis does not cover these, but we are willing to explore them in future work.
> >
> > > How to detect the accuracy of a triplet when dealing with open domain? How to deal with bias, etc.?
> > >
> >
> > If we understand the question correctly, the reviewer is asking how to really know whether a predicted triplet is a true/false answer, instead of resorting to the incomplete test data for a degraded metric. One most reliable way is to ask human evaluators for help, which might incur expensive cost. For example, reviewer Lj1a suggests a paper ([https://arxiv.org/abs/2108.01387](https://arxiv.org/abs/2108.01387)), which constructed such a dataset leveraging human effort. However, it is dealing with the triplet classification problem instead of link prediction, thus not directly applicable to our setting. As for how to deal with bias, we guess the bias can come from the KGC models, the disparate distribution of missing facts, and even the KG building process (either bias in text corpus or bias from human experts). These can all influence the behavior in the open-world setting. If we can identify the bias source, we can develop more specific solutions to deal with it.

---

> ### Author Response · Authors · 2022-08-08
> **We look forward to your reply**
>
> We thank reviewer LAYz again for the inspiring comments to help us improve the paper.
>
> In response to the comments, we address the main concerns as follows:
>
> 1. Convincingness of the proposed evaluation. Our main aim is to highlight the degradation and inconsistency problems and then point out their reason, the focus-on-top feature. As for our proposed evaluation, experiments show their robustness in our paper.
> 2. The rationality of the assumptions, especially for the binary classification and uniformly random break-ties. We justify them in three aspects:
>     1. The link-prediction task and the previously proposed models can be considered a binary classification task.
>     2. We added additional experiments by training ComplEx on two datasets and visualized the score distribution of the trained models. The distribution shows that the positive and negative part scores can be divided by a threshold, so our assumptions are reasonable.
>     3. Our experiments on the Family dataset do not need these assumptions and are consistent with our theoretical analysis.
> 3. The freshness, authority, and accuracy of evaluation. We believe it is a good question for the general KGC task. We are glad to add the analysis on them in future work.
> 4. The accuracy of a triplet when dealing with open domain and its bias. We believe human evaluation is a reliable method to evaluate the triplets. However, it might be impractical considering the scale of triplets involved in the link prediction task. For bias, we need to analyze the source of the bias and try to eliminate it separately.
>
> We valued the reviewer's feedback and made a great effort in writing the response. Since there is about 1 day left in the discussion phase, would you mind letting us know if our response addresses your concern? If you think there are still other issues, please kindly let us know. We are happy to follow up with you before the discussion phase ends.

---

### Official Review · Reviewer_DMNd · 2022-07-14

**Rating:** 7
**Confidence:** 3
**Soundness:** 4 excellent
**Presentation:** 3 good
**Contribution:** 4 excellent

**Summary:**

This paper discusses the gap between close-world and open-world assumptions, which are two major assumptions for KGs. Even though the missing links in KGs are uncertain in advance, the current ranking-based evaluation metric such as MRR and Hits@K heavily relies on the close-world assumption. It may cause a wrong conclusion when comparing KG completion models. The theoretical analysis through the expectation for the rank of entities based on the model strength shows that when the links in a KG are sparse, the model gets stronger the increase of the metric MRR will be less and less significant. Thus, in this case, there is a possibility that the evaluation result is severely affected by its variance. It may cause inconsistent comparison results between models. Empirical analysis of the artificially created dataset shows that the theoretical analysis is consistent with the observed results.

**Questions:**

- Based on your conclusion that large $l$ can reduce the variance problem in the ranking-based metrics, Hits@K is more robust than MRR since it is the same as purging entities with low $l$ in predicted results. Is this understanding correct? Clearing this point is quite beneficial for readers comparing KG models.
- Related to the first question, it seems that just using Hits@1 can avoid the problem. If so, Is there any problem with this choice?

**Limitations:**

- This paper reveals the unreliable behavior of the current ranking-based evaluation metrics. However, only increasing test queries are presented as a solution.

**Strengths And Weaknesses:**

Strengths:

- The stated problems are theoretically and empirically investigated.
- The information in this paper is essential for creating appropriate test sets in the future since sparse relationships in the test set may make it difficult to evaluate models correctly.

Weakness:

- Although this paper reveals the problem in the current raking-based evaluation metrics, its solution is not presented, excluding the increase of queries in testing.

---

> ### Author Response · Authors · 2022-08-02
> **Author Response**
>
> Thank you for your constructive comments. We address them as follows.
>
> > Weakness: its solution is not presented, excluding the increase of queries in testing.
> >
>
> Besides increasing test queries, we also recommend adding some modified rank-based metrics (e.g., $p$-MRR and $\log$-MRR), which are less focus-on-top and therefore, more robust under the open-world assumption. These metrics are discussed in Section 5 and verified in Figure 5(b).
>
> > Based on your conclusion that large $l$ can reduce the variance problem in the ranking-based metrics, Hits@K is more robust than MRR since it is the same as purging entities with low l in predicted results. Is this understanding correct? Clearing this point is quite beneficial for readers comparing KG models. Related to the first question, it seems that just using Hits@1 can avoid the problem. If so, Is there any problem with this choice?
> >
>
> Thanks for the question. The understanding is actually not correct. Firstly, in our analysis, $l$ is the actual strength of the learned model (modeled as the positive-negative classification accuracy) but not the score/ranking of individual entities. We aim to compare the actual strength $l$ for two different models when we only have access to some metrics evaluated on the open-world KG. The Hits@K (when $K$ is not too small) is indeed more robust than MRR, but the reason is that it is less focus-on-top, and the degradation of the gradient is not too severe. According to Corollary 4.3, a larger $K$ is more beneficial for relieving the degradation (note that $\Phi(K − 1)$ is monotonically increasing for $K$). Therefore, the Hits@1 is not a good choice here because it causes a too slight increase of metric w.r.t. actual model strength improvement.

---

### Meta-Review · Area_Chair_eofD · 2022-08-23

**Recommendation:** Accept
**Confidence:** Certain

**Metareview:**

This paper studies the evaluation of knowledge graph completion under an open-world assumption, where the training or test sets may include many unknown missing facts. It shows that the currently most-used metrics may not sufficiently reflect the true model performance and suggests alternative metrics to address this issue.  The reviewers found the problem this paper studies is fundamental, and well investigated both theoretically and empirically, which should benefit the research community in this field, although reviewers still have concerns about the assumption in this study and the lack of experiments on more realistic KGs. Overall the merits of the paper outweigh the drawbacks and an acceptance is recommended.

**Award:**

No

---

### Decision · Program_Chairs · 2022-09-14

Accept